# A Fast and Accurate Estimator for Large Scale Linear Model via Data Averaging

**Rui Wang**
Center for Applied Statistics
and School of Statistics
Renmin University of China
Beijing 100872, China 446100240@qq.com

**Yanyan Ouyang**
Center for Applied Statistics
and School of Statistics
Renmin University of China
Beijing 100872, China staoyyy@ruc.edu.cn

**Panpan Yu**
NavInfo
Beijing 100094, China
yupanpan@navinfo.com

**Wangli Xu**
Center for Applied Statistics
and School of Statistics
Renmin University of China
Beijing 100872, China wlxu@ruc.edu.cn

## Abstract

This work is concerned with the estimation problem of linear model when the sample size is extremely large and the data dimension can vary with the sample size. In this setting, the least square estimator based on the full data is not feasible with limited computational resources. Many existing methods for this problem are based on the sketching technique which uses the sketched data to perform least square estimation. We derive fine-grained lower bounds of the conditional mean squared error for sketching methods. For sampling methods, our lower bound provides an attainable optimal convergence rate. Our result implies that when the dimension is large, there is hardly a sampling method can have a faster convergence rate than the uniform sampling method. To achieve a better statistical performance, we propose a new sketching method based on data averaging. The proposed method reduces the original data to a few averaged observations. These averaged observations still satisfy the linear model and are used to estimate the regression coefficients. The asymptotic behavior of the proposed estimation procedure is studied. Our theoretical results show that the proposed method can achieve a faster convergence rate than the optimal convergence rate for sampling methods. Theoretical and numerical results show that the proposed estimator has good statistical performance as well as low computational cost.

## 1 Introduction

Linear regression model is one of the simplest and the most fundamental models in statistics and machine learning. Suppose one collects independent and identically distributed (i.i.d.) observations $\{Z_i, y_i\}_{i=1}^N$, where $Z_i \in \mathbb{R}^p$ is the vector of the predictors and $y_i \in \mathbb{R}$ is the response. The linear model assumes

$$y_i = \beta_0 + Z_i^\top \boldsymbol{\beta}_1 + \varepsilon_i, \quad i = 1, \dots, N, \tag{1}$$

where $\beta_0 \in \mathbb{R}$, $\boldsymbol{\beta}_1 \in \mathbb{R}^p$ are the unknown coefficients and $\varepsilon_1, \dots, \varepsilon_N$ are random variables representing noise. Let $X_i = (1, Z_i^\top)^\top$, $\boldsymbol{\beta} = (\beta_0, \boldsymbol{\beta}_1^\top)^\top$, $\mathbf{y} = (y_1, \dots, y_N)^\top$ and $\mathbf{X} = (X_1, \dots, X_N)^\top$. The classical least square estimator equals $\arg\min_{\boldsymbol{\beta} \in \mathbb{R}^{p+1}} \|\mathbf{y} - \mathbf{X}\boldsymbol{\beta}\|^2$. If $\mathbf{X}$ has full column rank, the least square estimator equals $(\sum_{i=1}^N X_i X_i^\top)^{-1} \sum_{i=1}^N X_i y_i$ and the direct computation costs $O(Np^2)$

37th Conference on Neural Information Processing Systems (NeurIPS 2023).

time. For large scale linear models with $N \gg p$, the computing time $O(Np^2)$ of the exact least square estimator is not negligible. Faster estimators of $\boldsymbol{\beta}$ can largely facilitate the practical data analysis pipelines.

Numerous research efforts have been devoted to the estimation problem for large scale linear model. Many existing work in this area can be understood as matrix sketching methods which explicitly or implicitly use matrix sketches as surrogates for the original observations to reduce the data size. Specifically, sketching methods solve the sketched least square problem

$$\min_{\boldsymbol{\beta} \in \mathbb{R}^{p+1}} \|\mathbf{O}^\top \mathbf{y} - \mathbf{O}^\top \mathbf{X} \boldsymbol{\beta}\|^2, \tag{2}$$

where $\mathbf{O} \in \mathbb{R}^{N \times n}$ is a sketching matrix with $n \ll N$. The solution to the problem (2) is the least square estimator based on the reduced data $\mathbf{O}^\top \mathbf{X} \in \mathbb{R}^{n \times (p+1)}$ and $\mathbf{O}^\top \mathbf{y} \in \mathbb{R}^n$. Since $n \ll N$, the sketched least square problem can be solved much faster than the least square estimator for the full data. In this paper, we only consider the case that $\mathbf{O}$ is independent of $\varepsilon_1, \ldots, \varepsilon_N$. That is, $\mathbf{O}$ may rely on $\mathbf{X}$, but not rely on $\mathbf{y}$. This guarantees that the solution to (2) is an unbiased estimator of $\boldsymbol{\beta}$. Note that sampling methods are special cases of the sketching framework (2). In fact, for a sampling method, each column of $\mathbf{O}$ is a vector whose elements are all 0 except one that equals 1. Sketching methods have been intensively researched in algorithmic aspect; see Mahoney [2010], Woodruff [2014], Drineas and Mahoney [2016] for reviews. Recently, the statistical aspect of sketching methods also draws much attention; see, e.g., Ma et al. [2015], Raskutti and Mahoney [2016], Wang et al. [2017], Dobriban and Liu [2019], Ma et al. [2020], Ahfock et al. [2021].

Probably the simplest sketching method is the uniform sampling method which randomly selects $n$ observations with equal probability to form the reduced data. Recently, Pilanci and Wainwright [2016] provides a minimax lower bound for the mean squared prediction error of random sketching methods. Theorem 1 of Pilanci and Wainwright [2016] shows that for a large class of random sketching methods, including many existing data-oblivious sketching methods and sampling methods, the convergence rate of the mean squared prediction error can not be faster than the uniform sampling method. Hence it is a nontrivial task to construct a sketching method which has significantly better statistical performance than the uniform sampling method.

Recently, Wang et al. [2019] initiates the study of sampling methods based on extreme values. Motivated by the D-optimal criterion, Wang et al. [2019] proposed the information-based optimal subdata selection (IBOSS) algorithm which successively selects informative observations based on extreme values of variables. They showed that for fixed $p$, the estimator produced by the IBOSS algorithm can have a faster convergence rate than the uniform sampling method. Meanwhile, the computation of the IBOSS algorithm can be completed within $O(Np + np^2)$ time which has the same order as the uniform sampling method if $n = cN/p$ for some constant $c > 0$. Now the algorithm of Wang et al. [2019] has become the building block of some recent methods for large scale problems. For example, Wang [2019] proposed an algorithm which combines the algorithm of Wang et al. [2019] and the divide and conquer strategy. Cheng et al. [2020] extended the algorithm of Wang et al. [2019] to the logistic regression model. Existing asymptotic results for the IBOSS algorithm are obtained in the setting of fixed $n$ and $p$. At present, there is still a lack of theoretical understanding of the behavior of the IBOSS algorithm in the setting of varying $n$ and $p$.

The IBOSS algorithm is a sampling method and is therefore an instance of the sketching framework (2). Interestingly, the IBOSS algorithm can surpass the minimax lower bound of Pilanci and Wainwright [2016]. In fact, a key condition for Theorem 1 of Pilanci and Wainwright [2016] does not hold for the IBOSS algorithm. Thus, the IBOSS algorithm is not restricted by the minimax bound of Pilanci and Wainwright [2016]. This fact is detailed in Section 2. Note that there are many potential sketching methods which are not restricted by the minimax bound of Pilanci and Wainwright [2016]. To give a more comprehensive understanding of the behavior of these sketching methods, we derive fine-grained lower bounds for the conditional mean squared error of the sketched least square estimators produced by (2) in the setting that $Z_i$ is a standard normal random vector. In particular, our result provides a lower bound for any sampling method which may possibly rely on $\mathbf{X}$ but does not rely on $\mathbf{y}$. It turns out that if $p \ll \log(N/n)$, then the optimal lower bound for sampling methods can have a faster convergence rate than the uniform sampling method. On the other hand, if $\log(N/n) \ll p$, any sampling method can not largely surpass the uniform sampling method. Furthermore, we derive the asymptotic behavior of the IBOSS algorithm in the setting of varying $n$ and $p$. It turns out that under certain conditions, the IBOSS algorithm can achieve the optimal rate for sampling methods.

Table 1: Theoretical performance of the ideal sampling method (abbreviated as ISM), the IBOSS algorithm and the proposed method when $Z_1 \sim \mathcal{N}(\mathbf{0}_p, \mathbf{I}_p)$. Assume that as $N \to \infty$, $p \to \infty$, $p^3(\log(p))^4 \log(N)/N \to \infty$, $n = O(N^\epsilon)$ for some $0 < \epsilon < 1/2$ and $p = O(n^{1/2-\epsilon^*})$ for some $0 < \epsilon^* < 1/2$, $\log(N/n) = O(p^2)$. See Theorems 2, 3, 4. The reported computing time is under the assumption that the multiplication of an $m \times n$ matrix and an $n \times p$ matrix costs $O(mnp)$ time, and the inversion of a $p \times p$ matrix costs $O(p^3)$ time.

| Methods | Reduced sample size | $\mathrm{E}\left\{\|\hat{\boldsymbol{\beta}}_{\mathrm{I}} - \boldsymbol{\beta}\|^2 \mid \mathbf{Z}\right\}$ | Computing time |
|---------|---------------------|-----------|----------------|
| ISM | $n$ | $O_P\left(\frac{p^2}{n\left(p+\log\left(\frac{N}{n}\right)\right)}\right)$ | — |
| IBOSS | $n$ | $O_P\left(\frac{p^2}{n\left(p+\log\left(\frac{N}{n}\right)\right)}\right)$ | $O(Np + np^2)$ |
| NEW | $2p$ | $(1 + o_P(1))\frac{p^2\sigma_\varepsilon^2}{2\log(2p)N}$ | $O(Np + p^3)$ |

For large scale linear models, it is often the case that $\log(N/n) \ll p$. In this case, any sampling method can not have a significantly better statistical performance than the uniform sampling method. Inspired by this phenomenon, we propose an alternative sketching method which can reduce the full data to just a few observations while the resulting estimator of $\boldsymbol{\beta}$ may have smaller conditional mean squared error than sampling methods. The proposed method is based on data averaging. The main idea is to partition the observations into $2p$ groups such that the averages of $Z_i$ within groups are separated. The least square estimator based on $2p$ averaged observations is used to estimate $\boldsymbol{\beta}$. The computation of the proposed method can be completed within $O(Np + p^3)$ time. Our theoretical results show that the proposed method can have a faster convergence rate than any sampling methods with comparative computing time. Also, the proposed method reduces the full data to merely $2p$ averaged observations. These averaged observations also satisfy the linear model (1) and have independent errors. Consequently, it is convenient to further compute other estimators or conduct statistical inferences using the reduced data. The good performance of the proposed estimator is also verified by simulation results and a real data example. Table 1 summarizes the theoretical performance of the proposed method and compare it with the ideal sampling method implied by Theorem 2 and the IBOSS algorithm.

The rest of the paper is organized as follows. Section 2 investigates lower bounds for the conditional mean squared error of the sketched least square estimators produced by (2). In Section 3, we propose a data averaging method to estimate $\boldsymbol{\beta}$ and investigate its asymptotic behavior. Section 4 presents the simulation results briefly. Section 5 concludes the paper. The simulation results, a real data analysis and all proofs are deferred to the Supplement Material.

We close this section by introducing some notations and assumptions that will be used throughout the paper. For any real number $w$, let $\lfloor w \rfloor$ denote the largest integer not larger than $w$. For any vector $W$, let $\|W\|$ denote the Euclidean norm of $W$. For any matrix $\mathbf{B}$, let $\|\mathbf{B}\|$ and $\|\mathbf{B}\|_F$ denote the operator norm and the Frobenious norm of $\mathbf{B}$, respectively. Moreover, denote by $\mathbf{B}_{:,j}$ the $j$th column of $\mathbf{B}$. If $\mathbf{B}$ is symmetric, denote by $\lambda_i(\mathbf{B})$ the $i$th largest eigenvalue of $\mathbf{B}$. In this paper, the symmetric matrices are equipped with Loewner partial order. That is, for two symmetric matrices $\mathbf{B}_1$ and $\mathbf{B}_2$, $\mathbf{B}_1 > \mathbf{B}_2$ if and only if $\mathbf{B}_1 - \mathbf{B}_2$ is positive definite. For a positive semidefinite matrix $\mathbf{B}$, let $\mathbf{B}^{1/2}$ denote a positive semidefinite matrix such that $(\mathbf{B}^{1/2})^2 = \mathbf{B}$. For any set $\mathscr{A}$, denote by $\mathscr{A}^{\complement}$ its complement and $\mathrm{Card}(\mathscr{A})$ its cardinality. Let $\Phi(x)$ and $\varphi(x)$ denote the distribution function and density function of the standard normal distribution, respectively. For random variables $\xi \in \mathbb{R}$ and $\eta > 0$, $\xi = o_P(\eta)$ means that $\xi/\eta$ converges to 0 in probability, and $\xi = O_P(\eta)$ means that $\xi/\eta$ is bounded in probability.

Let $N$ denote the size of full sample, $p$ denote the dimension of covariates. Let $\mathbf{Z} = (Z_1, \ldots, Z_N)^\top$ be an $N \times p$ matrix of covariates. Denote by $z_{i,j}$ the $j$th element of $Z_i$, $i = 1, \ldots, N$, $j = 1, \ldots, p$. Let $z_{(1),j} \leq \cdots \leq z_{(N),j}$ denote the order statistics of $\{z_{i,j}\}_{i=1}^N$, $j = 1, \ldots, p$. The following assumption for the data distribution is assumed throughout the paper.

**Assumption 1** *Suppose $\{Z_i, y_i\}_{i=1}^N$ are i.i.d. and satisfy the linear model* (1)*, where* $\mathrm{E}(\varepsilon_1) = 0$, $\mathrm{Var}(\varepsilon_1) = \sigma_\varepsilon^2$ *and* $\varepsilon_1$ *is independent of* $Z_1$. *Suppose* $Z_1$ *has a density function* $f(z_{1,1}, \ldots, z_{1,p})$ *with respect to the Lebesgue measure on* $\mathbb{R}^p$, *and* $\mathrm{E}(Z_1) = \mu$, $\mathrm{Cov}(Z_1) = \Sigma$ *are finite. Suppose* $\Sigma = (\sigma_{i,j})_{i,j=1}^p$ *is positive definite. Suppose* $r > 0$. *As* $N \to \infty$, *the dimension* $p$ *is a function of* $N$, *while the distribution of* $(Z_1, y_1)$ *only relies on* $p$. *Finally, assume* $\sigma_\varepsilon^2$ *is a constant which does not depend on* $N$.

For simplicity, our notations suppress the dependence of $p$ on $N$, and the dependence of the distribution of $(Z_1, y_1)$ on $p$.

## 2 Risk bounds for sketched least square estimators

Theorem 1 of Pilanci and Wainwright [2016] provides a minimax lower bound for the mean squared prediction error of random sketching methods. Their result implies that under certain conditions, there exists a constant $C > 0$ such that for any estimator $\hat{\beta}$ which only relies on $(\mathbf{O}^\top \mathbf{X}, \mathbf{O}^\top \mathbf{y})$,

$$\sup_{\beta \in \mathbb{R}^p} \mathrm{E}\{N^{-1} \|\mathbf{X}(\hat{\beta} - \beta)\|^2 \mid \mathbf{Z}\} \geq \frac{Cp}{n} \sigma_\varepsilon^2.$$

The optimal convergence rate $p/n$ can be achieved by the least square estimator based on $n$ uniformly selected observations. A key condition for the above result is that

$$\| \mathrm{E}(\mathbf{O}(\mathbf{O}^\top \mathbf{O})^{-1} \mathbf{O}^\top \mid \mathbf{Z})\| \leq cn/N \tag{3}$$

for some constant $c > 0$.

The result of Pilanci and Wainwright [2016] can be applied to general estimators based on $(\mathbf{O}^\top \mathbf{X}, \mathbf{O}^\top \mathbf{y})$. In this paper, however, we focus on the least square estimator based on $(\mathbf{O}^\top \mathbf{X}, \mathbf{O}^\top \mathbf{y})$. Let $\hat{\beta}_{\mathbf{O}}$ denote the solution to the sketched least square problem (2). In this paper, we use the conditional mean squared error $\mathrm{E}\{\|\hat{\beta}_{\mathbf{O}} - \beta\|^2 \mid \mathbf{Z}\}$ to measure the performance of $\hat{\beta}_{\mathbf{O}}$. The following theorem gives a lower bound for the conditional mean squared error of $\hat{\beta}_{\mathbf{O}}$.

**Theorem 1** *Suppose Assumption 1 holds,* $Z \sim \mathcal{N}(\mathbf{0}_p, \mathbf{I}_p)$, *the sketching matrix* $\mathbf{O}$ *is an* $N \times n$ *matrix with full column rank. Assume that* $\mathbf{O}$ *is independent of* $\varepsilon_1, \ldots, \varepsilon_N$ *and with probability* 1, $\mathbf{O}^\top \mathbf{X}$ *has full column rank. Suppose as* $N \to \infty$, $p/N \to 0$. *Then as* $N \to \infty$,

$$\mathrm{E}\left\{\|\hat{\beta}_{\mathbf{O}} - \beta\|^2 \mid \mathbf{Z}\right\} \geq (1 + o_P(1)) \left\|\mathrm{E}(\mathbf{O}(\mathbf{O}^\top \mathbf{O})^{-1} \mathbf{O}^\top \mid \mathbf{Z})\right\|^{-1} \frac{p+1}{N} \sigma_\varepsilon^2.$$

Theorem 1 gives an explicit characterization of the impact of $\mathrm{E}(\mathbf{O}(\mathbf{O}^\top \mathbf{O})^{-1} \mathbf{O}^\top \mid \mathbf{Z})$ on the lower bound of $\mathrm{E}\{\|\hat{\beta}_{\mathbf{O}} - \beta\|^2 \mid \mathbf{Z}\}$. Pilanci and Wainwright [2016] showed that the condition (3) is satisfied by many classical sketching methods. Under the conditions of Theorem 1, for sketching methods satisfying the condition (3), the convergence rate of $\mathrm{E}\{\|\hat{\beta}_{\mathbf{O}} - \beta\|^2 \mid \mathbf{Z}\}$ is lower bounded by $p/n$, which is the convergence rate for the uniform sampling method. Thus, in order to achieve a faster convergence rate than the uniform sampling method, the condition (3) should be violated.

Many existing sketching methods are through sampling the observations. For sampling methods, $\mathbf{O}$ is a column orthogonal matrix and each column of $\mathbf{O}$ has a single nonzero element with value 1. Hence $\mathbf{O}(\mathbf{O}^\top \mathbf{O})^{-1} \mathbf{O}^\top$ is a diagonal matrix whose diagonal elements are zeros and ones. For the IBOSS algorithm of Wang et al. [2019], the selected observations are completely determined by $\mathbf{X}$ and does not rely on additional randomness. Consequently $\| \mathrm{E}(\mathbf{O}(\mathbf{O}^\top \mathbf{O})^{-1} \mathbf{O}^\top \mid \mathbf{Z})\| = \|\mathbf{O}(\mathbf{O}^\top \mathbf{O})^{-1} \mathbf{O}^\top\| = 1$. In this case, the lower bound provided by Theorem 1 has rate $p/N$ which is too loose. The following theorem gives a tighter lower bound of the mean squared error for sampling methods.

**Theorem 2** *Suppose Assumption 1 holds,* $Z \sim \mathcal{N}(\mathbf{0}_p, \mathbf{I}_p)$, *the sketching matrix* $\mathbf{O}$ *is an* $N \times n$ *matrix with full column rank. Assume that* $\mathbf{O}$ *is independent of* $\varepsilon_1, \ldots, \varepsilon_N$ *and with probability* 1, $\mathbf{O}^\top \mathbf{X}$ *has full column rank. Furthermore, suppose* $\mathrm{E}(\mathbf{O}(\mathbf{O}^\top \mathbf{O})^{-1} \mathbf{O}^\top \mid \mathbf{Z}) = \mathrm{diag}(d_1, \ldots, d_N)$. *Let* $d_{\max} = \max_{i \in \{1, \ldots, N\}} d_i$ *Then*

$$\mathrm{E}\left\{\|\hat{\beta}_{\mathbf{O}} - \beta\|^2 \mid \mathbf{Z}\right\} \geq \frac{p^2}{6n \left(p + \log\left(\frac{Nd_{\max}}{n}\right)\right) + O_P(n)} \sigma_\varepsilon^2.$$

If the matrix $\mathrm{E}(\mathbf{O}(\mathbf{O}^\top\mathbf{O})^{-1}\mathbf{O}^\top \mid \mathbf{Z}) = \mathrm{diag}(d_1,\ldots,d_N)$ is diagonal, then

$$d_{\max} = \max_{\alpha\in\mathbb{R}^N,\|\alpha\|=1} \alpha^\top \mathrm{E}(\mathbf{O}(\mathbf{O}^\top\mathbf{O})^{-1}\mathbf{O}^\top \mid \mathbf{Z})\alpha \le \mathrm{E}(\max_{\alpha\in\mathbb{R}^N,\|\alpha\|=1} \alpha^\top \mathbf{O}(\mathbf{O}^\top\mathbf{O})^{-1}\mathbf{O}^\top\alpha \mid \mathbf{Z}) = 1.$$

Under the conditions of Theorem 2, the optimal convergence rate for sampling methods is lower bounded by

$$\frac{p^2}{n\left(p + \log\left(\frac{N}{n}\right)\right)}. \tag{4}$$

Note that if $p \ll \log(N/n)$, then the rate (4) is faster than the uniform sampling method. Theorem 4 in Section 3.2 will show that under certain conditions, the method of Wang et al. [2019] can achieve the optimal rate (4).

It is worth mentioning that Theorems 1 and 2 are obtained under the condition $Z_i \sim \mathcal{N}(\mathbf{0}_p, \mathbf{I}_p)$. Perhaps, these results can be extended to the case that $Z_i$ has a general multivariate distribution. However, such results may not be valid if the distribution of $Z_i$ has heavier tail than normal distribution. In fact, our numerical results imply that a faster convergence rate may be achieved when the distribution of $Z_i$ has heavy tail.

## 3 An estimator via data averaging

In this section, we would like to propose a new sketching method which can hopefully have good statistical performance with low computational cost. To be simple, when considering computation time, it is understood that the multiplication of an $m \times n$ matrix and $n \times p$ matrix costs $O(mnp)$ time, and the inversion of a $p \times p$ matrix costs $O(p^3)$ time. Note that the computation time $O(Np)$ is essential if each observation is accessed at least once, e.g., to be loaded into the memory. The sketched least square problem (2) involves $n$ reduced observations and the direct computation costs $O(np^2 + p^3)$ computation time. The computation time $O(p^3)$ comes from the inversion of a $(p+1) \times (p+1)$ matrix which is essential no matter how $n$ is chosen. Hence the direct computation of any reasonable estimator which uses the information of the full data requires at least $O(Np + p^3)$ time. Thus, we restrict our attention to algorithms that can be completed within $O(Np + p^3)$ time.

To complete the computation within $O(Np + p^3)$ time, one needs to take $n = O(N/p + p)$. Theorem 2 implies that if $n = c_1 N/p + c_2 p$ where $c_1, c_2 > 0$ are constants, then the optimal convergence rate for sampling methods reduces to $p/n$ which is equal to the convergence rate for the uniform sampling method. Also, for large $N$, the reduced sample size $n = c_1 N/p + c_2 p$ may still be large. To achieve a faster convergence rate and a better reduction of data, we would like to consider sketching methods other than sampling methods. This motivates us to propose a new data averaging method.

### 3.1 Methodology

Let $\mathscr{J}_1,\ldots,\mathscr{J}_k \subset \{1,\ldots,N\}$ be $k$ mutually disjoint index sets, each containing $r$ indices, and $\bigcup_{i=1}^k \mathscr{J}_i = \{1,\ldots,N\}$. To use the information of the full data, we assume $N = kr$. Let $\bar{Z}_j = r^{-1}\sum_{i\in\mathscr{J}_j} Z_i$ and $\bar{y}_j = r^{-1}\sum_{i\in\mathscr{J}_j} y_i$ be the averaged observation within the $j$th index set, $j = 1,\ldots,k$. It can be seen that

$$\bar{y}_j = \beta_0 + \bar{Z}_j^\top \boldsymbol{\beta}_1 + \bar{\varepsilon}_j,$$

where $\bar{\varepsilon}_j = r^{-1}\sum_{i\in\mathscr{J}_j} \varepsilon_i$. Suppose that the choice of the index sets $\mathscr{J}_1,\ldots,\mathscr{J}_k$ is based on the covariates $\{Z_i\}_{i=1}^N$ and does not rely on the responses $\{y_i\}_{i=1}^N$. Then $\bar{\varepsilon}_1,\ldots,\bar{\varepsilon}_k$ are mutually independent and are independent of $\{\bar{Z}_j\}_{j=1}^k$. Also, $\bar{\varepsilon}_j$ has mean 0 and variance $\sigma_\varepsilon^2/r$. Thus, the averaged observations also satisfy the linear model and one can estimate $\boldsymbol{\beta}$ by the least square estimator base on $k$ reduced observations as $\hat{\boldsymbol{\beta}} = (\sum_{j=1}^k \bar{X}_j\bar{X}_j^\top)^{-1}(\sum_{j=1}^k \bar{X}_j\bar{y}_j)$, where $\bar{X}_j = r^{-1}\sum_{i\in\mathscr{J}_j} X_i$, $j = 1,\ldots,k$. We would like to choose $\mathscr{J}_1,\ldots,\mathscr{J}_k$ such that $\hat{\boldsymbol{\beta}}$ is a fast and accurate estimator of $\boldsymbol{\beta}$. Let $\mathbf{H} = \sum_{\ell=1}^k (\bar{Z}_\ell - \bar{Z})(\bar{Z}_\ell - \bar{Z})^\top$ and $\bar{Z} = N^{-1}\sum_{i=1}^N Z_i$. The

conditional mean squared error of $\hat{\boldsymbol{\beta}}$ is

$$
\begin{aligned}
\mathrm{E}(\|\hat{\boldsymbol{\beta}} - \boldsymbol{\beta}\|^2 \mid \mathbf{Z}) &= \frac{k\sigma_\epsilon^2}{N} \operatorname{tr}\left\{ \begin{pmatrix} k & \sum_{j=1}^k \bar{Z}_j^\top \\ \sum_{j=1}^k \bar{Z}_j & \sum_{j=1}^k \bar{Z}_j \bar{Z}_j^\top \end{pmatrix}^{-1} \right\} \\
&= \frac{k\sigma_\epsilon^2}{N} \operatorname{tr}\left\{ \begin{pmatrix} \frac{1}{k} + \bar{Z}^\top \mathbf{H}^{-1} \bar{Z} & -\bar{Z}^\top \mathbf{H}^{-1} \\ -\mathbf{H}^{-1} \bar{Z} & \mathbf{H}^{-1} \end{pmatrix} \right\} \\
&= \left( k\left( \operatorname{tr}(\mathbf{H}^{-1}) + \bar{Z}^\top \mathbf{H}^{-1} \bar{Z} \right) + 1 \right) \frac{\sigma_\varepsilon^2}{N}.
\end{aligned}
$$

In order to achieve a good statistical accuracy, we would like to choose the index sets such that $\operatorname{tr}(\mathbf{H}^{-1}) + \bar{Z}^\top \mathbf{H}^{-1} \bar{Z}$ is minimized.

First we consider the simplest case of $p = 1$. In this case, the matrix $\mathbf{H}$ reduces to a real number. Since $\boldsymbol{\beta} \in \mathbb{R}^2$, one needs at least two observations to estimate $\boldsymbol{\beta}$. To achieve maximum reduction of data, we take $k = 2$. Then $\mathbf{H}$ takes its maximum when $\mathscr{J}_1 = \{i \in \{1, \ldots, N\} : z_{i,1} \leq z_{(N/2),1}\}$ and $\mathscr{J}_2 = \{i \in \{1, \ldots, N\} : z_{i,1} \geq z_{(N/2+1),1}\}$. The least square estimator of $\boldsymbol{\beta}$ based on the averaged observations is $\hat{\boldsymbol{\beta}} = \left( (\bar{Z}_2 \bar{y}_1 - \bar{Z}_1 \bar{y}_2)/(\bar{Z}_2 - \bar{Z}_1), (\bar{y}_2 - \bar{y}_1)/(\bar{Z}_2 - \bar{Z}_1) \right)^\top$. The above estimator $(\bar{y}_2 - \bar{y}_1)/(\bar{Z}_2 - \bar{Z}_1)$ of $\boldsymbol{\beta}_1$ is considered in Barton and Casley [1958] as a quick estimate of $\boldsymbol{\beta}_1$, which only considered the case of $p = 1$. To the best of our knowledge, no previous study generalized this estimator of Barton and Casley [1958] to the case $p > 1$.

For the general case of $p \geq 1$, the exact minimizer of $\mathrm{E}(\|\hat{\boldsymbol{\beta}} - \boldsymbol{\beta}\|^2 \mid \mathbf{Z})$ may not be easy to obtain. A simpler criterion to choose the index sets is to maximize the trace $\operatorname{tr}(\mathbf{H}) = \sum_{\ell=1}^k \|\bar{Z}_\ell - \bar{Z}\|^2$. This problem is equivalent to minimizing $\sum_{\ell=1}^k \sum_{i \in \mathscr{J}_\ell} \|Z_i - \bar{Z}_\ell\|^2$ and is an instance of the balanced $k$-means clustering problem; see Lin et al. [2019] and the references therein. Unfortunately, algorithms for the $k$-means clustering problem are computationally intensive. In fact, for the vanilla $k$-means algorithm, each iteration takes $O(Npk)$ time which even exceeds the computing time of the least square estimator based on the full data. To achieve a balance between the statistical accuracy and the computing time, we deal with each variable in turn. We take $k = 2p$ and for $j = 1, \ldots, p$, we determine two index sets, namely $\mathscr{L}_{r,j}$ and $\mathscr{R}_{r,j}$, based on the $j$th variable. Hence the set $\{1, \ldots, N\}$ is partitioned into $2p$ index sets $\mathscr{L}_{r,1}, \ldots, \mathscr{L}_{r,p}$ and $\mathscr{R}_{r,1}, \ldots, \mathscr{R}_{r,p}$, each containing $r = N/(2p)$ indices. The choice of these index sets is based on the following lower bound of $\operatorname{tr}(\mathbf{H})$,

$$
\begin{aligned}
\operatorname{tr}(\mathbf{H}) &= \sum_{j=1}^p \sum_{\ell=1}^p \left\{ \left( \frac{1}{r} \sum_{i \in \mathscr{L}_{r,\ell}} z_{i,j} - \frac{1}{N} \sum_{i=1}^N z_{i,j} \right)^2 + \left( \frac{1}{r} \sum_{i \in \mathscr{R}_{r,\ell}} z_{i,j} - \frac{1}{N} \sum_{i=1}^N z_{i,j} \right)^2 \right\} \\
&\geq \sum_{j=1}^p \left[ \sum_{\ell=1}^{j-1} \left\{ \left( \frac{1}{r} \sum_{i \in \mathscr{L}_{r,\ell}} z_{i,j} - \frac{1}{N} \sum_{i=1}^N z_{i,j} \right)^2 + \left( \frac{1}{r} \sum_{i \in \mathscr{R}_{r,\ell}} z_{i,j} - \frac{1}{N} \sum_{i=1}^N z_{i,j} \right)^2 \right\} \right. \\
&\quad \left. + \left\{ \left( \max(\tilde{z}_j - \frac{1}{r} \sum_{i \in \mathscr{L}_{r,j}} z_{i,j}, 0) \right)^2 + \left( \max(\frac{1}{r} \sum_{i \in \mathscr{R}_{r,j}} z_{i,j} - \tilde{z}_j, 0) \right)^2 \right\} \right],
\end{aligned}
$$

where $\tilde{z}_j = (2r(p - j + 1))^{-1} \sum_{i \notin \bigcup_{\ell=1}^{j-1}(\mathscr{L}_{r,\ell} \cup \mathscr{R}_{r,\ell})} z_{i,j}$. For $j = 1, \ldots, p$, we choose $\mathscr{L}_{r,j}$ and $\mathscr{R}_{r,j}$ to maximize the $j$th term of the above lower bound. Specifically, the first term of the above lower bound is $\{\max(\tilde{z}_1 - \sum_{i \in \mathscr{L}_{r,1}} z_{i,1}/r, 0)\}^2 + \{\max(\sum_{i \in \mathscr{R}_{r,1}} z_{i,1}/r - \tilde{z}_1, 0)\}^2$, which takes its maximum when $\mathscr{L}_{r,1} = \{i \in \{1, \ldots, N\} : z_{i,1} \leq \gamma_{1,1}\}$ and $\mathscr{R}_{r,1} = \{i \in \{1, \ldots, N\} : z_{i,1} \geq \gamma_{2,1}\}$ where $\gamma_{1,1} = z_{(r),1}$ and $\gamma_{2,1} = z_{(N-r+1),1}$. After obtaining the index sets $\mathscr{L}_{r,1}, \ldots, \mathscr{L}_{r,j-1}$ and $\mathscr{R}_{r,1}, \ldots, \mathscr{R}_{r,j-1}$, we choose $\mathscr{L}_{r,j}$ and $\mathscr{R}_{r,j}$ to maximize the $j$th term of the above lower bound, which is equivalent to maximizing $\{\max(\tilde{z}_j - \sum_{i \in \mathscr{L}_{r,j}} z_{i,j}/r, 0)\}^2 + \{\max(\sum_{i \in \mathscr{R}_{r,j}} z_{i,j}/r - \tilde{z}_j, 0)\}^2$. Hence we choose $\mathscr{L}_{r,j}$ and $\mathscr{R}_{r,j}$ to be the indices of the remaining observations whose $j$th variable is no larger than $\gamma_{1,j}$ and no less than $\gamma_{2,j}$, respectively, where $\gamma_{1,j}$ and $\gamma_{2,j}$ are the $r$th smallest and the $r$th largest element of $\{z_{i,j} : i \in \{1, \ldots, N\} \backslash (\bigcup_{\ell=1}^{j-1}(\mathscr{L}_{r,\ell} \cup \mathscr{R}_{r,\ell}))\}$, respectively. We average the observations within the groups $\mathscr{L}_{r,1}, \ldots, \mathscr{L}_{r,p}$ and $\mathscr{R}_{r,1}, \ldots, \mathscr{R}_{r,p}$. Finally, we use the least square estimator based on the $2p$ averaged observations to estimate $\boldsymbol{\beta}$. The proposed estimation procedure is summarized in Algorithm 1.

---

**Algorithm 1:** Data averaging algorithm

---

**Input:** Observations $\{Z_i, y_i\}_{i=1}^N$, covariate dimension $p$
**Output:** Estimator of $\boldsymbol{\beta}$
$r = \frac{N}{2p}$ is assumed to be an integer
**for** $j \in \{1, ..., p\}$ **do**
    $\gamma_{1,j} \leftarrow$ the $r$th smallest element of $\{z_{i,j} : i \in \{1, \ldots, N\} \backslash (\bigcup_{\ell=1}^{j-1}(\mathscr{L}_{r,\ell} \cup \mathscr{R}_{r,\ell}))\}$
    $\gamma_{2,j} \leftarrow$ the $r$th largest element of $\{z_{i,j} : i \in \{1, \ldots, N\} \backslash (\bigcup_{\ell=1}^{j-1}(\mathscr{L}_{r,\ell} \cup \mathscr{R}_{r,\ell}))\}$
    $\mathscr{L}_{r,j} \leftarrow \{i \in \{1, \ldots, N\} \backslash (\bigcup_{\ell=1}^{j-1}(\mathscr{L}_{r,\ell} \cup \mathscr{R}_{r,\ell})) : z_{i,j} \leq \gamma_{1,j}\}$
    $\mathscr{R}_{r,j} \leftarrow \{i \in \{1, \ldots, N\} \backslash (\bigcup_{\ell=1}^{j-1}(\mathscr{L}_{r,\ell} \cup \mathscr{R}_{r,\ell})) : z_{i,j} \geq \gamma_{2,j}\}$
    $\bar{Z}_j^L \leftarrow r^{-1} \sum_{i \in \mathscr{L}_j} Z_i, \quad \bar{X}_j^L = (1, \bar{Z}_j^{L\top})^\top, \quad \bar{y}_j^L \leftarrow r^{-1} \sum_{i \in \mathscr{L}_j} y_i$
    $\bar{Z}_j^R \leftarrow r^{-1} \sum_{i \in \mathscr{R}_j} Z_i, \quad \bar{X}_j^R = (1, \bar{Z}_j^{R\top})^\top, \quad \bar{y}_j^R \leftarrow r^{-1} \sum_{i \in \mathscr{R}_j} y_i$
$\hat{\boldsymbol{\beta}}_{\mathrm{A}} \leftarrow \left(\sum_{j=1}^p \bar{X}_j^L \bar{X}_j^{L\top} + \sum_{j=1}^p \bar{X}_j^R \bar{X}_j^{R\top}\right)^{-1} \left(\sum_{j=1}^p \bar{X}_j^L \bar{y}_j^L + \sum_{j=1}^p \bar{X}_j^R \bar{y}_j^R\right)$
**return** $\hat{\boldsymbol{\beta}}_{\mathrm{A}}$

---

In Algorithm 1, our strategy to select the index sets $\mathscr{L}_{r,j}$ and $\mathscr{R}_{r,j}$ is closely related to the IBOSS algorithm of Wang et al. [2019]. In fact, the index sets in Algorithm 1 is exactly the index sets selected by IBOSS algorithm with subdata size $n := N$. Of course, for IBOSS algorithm, taking $n = N$ is unreasonable since the sample size is not reduced. In fact, for IBOSS algorithm, one needs to take $n = O(N/p + p)$ to complete the computation within $O(Np + p^3)$ time. Thus, the selection procedures of the proposed method and the IBOSS algorithm have different behavior. Theorem 4 will show that under certain conditions, IBOSS can achieve the optimal convergence rate (4) among all sampling methods. We shall see that the statistical performance of Algorithm 1 is even better than the IBOSS algorithm.

Now we give an analysis of the computing time of Algorithm 1. Note that $\gamma_{1,j}$ and $\gamma_{2,j}$ are order statistics of no more than $N$ elements. It is known that the selection of an order statistic among $m$ elements can be completed within $O(m)$ time even in the worst case; see Paterson [1996]. Hence the computation of $\gamma_{1,1}, \ldots, \gamma_{1,p}$ and $\gamma_{2,1}, \ldots, \gamma_{2,p}$ can be completed within $O(Np)$ time in total. It takes only one scan of the full data to compute the averaged observations, which takes $O(Np)$ time. Finally, the computation of $\hat{\boldsymbol{\beta}}_{\mathrm{A}}$ based on $2p$ averaged observations can be completed within $O(p^3)$ time. In summary, Algorithm 1 can be completed within $O(Np + p^3)$ time and reduces the full data to merely $2p$ observations.

## 3.2 Asymptotic results

Now we investigate the asymptotic behavior of the conditional mean squared error of $\hat{\boldsymbol{\beta}}_{\mathrm{A}}$. In our asymptotic results, we treat $p$ as a function of $N$, and $N$ tends to infinity. Let $Z = (z_1, \ldots, z_p)^\top$ be a random vector which is independent of $\mathbf{Z}$ and $\mathbf{y}$ and has the same distribution as $Z_1$. The following theorem gives the exact limit of $\mathrm{E}\{\|\hat{\boldsymbol{\beta}}_{\mathrm{A}} - \boldsymbol{\beta}\|^2 \mid \mathbf{Z}\}$ when $Z$ is a standard normal random vector.

**Theorem 3** *Suppose that Assumption 1 holds, $r = N/(2p)$ is an integer, $N > 2p^2$, and $Z \sim \mathcal{N}(\mathbf{0}_p, \mathbf{I}_p)$. Also suppose that as $N \to \infty$, $p \to \infty$ and $p^3(\log(p))^4 \log(N)/N \to 0$. Then as $N \to \infty$,*

$$\mathrm{E}\left\{\|\hat{\boldsymbol{\beta}}_{\mathrm{A}} - \boldsymbol{\beta}\|^2 \mid \mathbf{Z}\right\} = (1 + o_P(1)) \frac{p^2 \sigma_\varepsilon^2}{2\log(2p)N}.$$

Theorem 3 implies that when $Z$ is a standard normal random vector, the conditional mean squared error of $\hat{\boldsymbol{\beta}}_{\mathrm{A}}$ has convergence rate $p^2/(\log(2p)N)$. On the other hand, for sampling methods with $n = c_1 N/p + c_2 p$ for constants $c_1, c_2 > 0$ such that the computing time may be comparable, Theorem 2 implies that the optimal convergence rate of the conditional mean squared error is $p^2/N$. In this view, the convergence rate of the proposed estimator is faster than sampling methods for $p \to \infty$.

The proposed algorithm is closely related to the IBOSS algorithm. We would like to derive the asymptotic behavior of the conditional mean squared error of $\hat{\boldsymbol{\beta}}_{\mathrm{I}}$. Theorem 6(i) of Wang et al. [2019] gives an asymptotic expression of the conditional covariance of $\hat{\boldsymbol{\beta}}_{\mathrm{I}}$ under the assumption that $Z$ is

normally distributed. It is implied that if $n$ and $p$ are fixed, then as $N \to \infty$,

$$\mathrm{E}\left\{\|\hat{\boldsymbol{\beta}}_{\mathrm{I}} - \boldsymbol{\beta}\|^2 \mid \mathbf{Z}\right\} = (1 + o_P(1))\left(\frac{p}{2\log(N)}\,\mathrm{tr}(\boldsymbol{\Sigma}^{-1}\,\mathrm{diag}(\boldsymbol{\Sigma})\boldsymbol{\Sigma}^{-1}) + 1\right)\frac{\sigma_\varepsilon^2}{n}. \tag{5}$$

Now we derive the fine-grained limiting behavior of $\mathrm{E}\left\{\|\hat{\boldsymbol{\beta}}_{\mathrm{I}} - \boldsymbol{\beta}\|^2 \mid \mathbf{Z}\right\}$ for varying $n$ and $p$. Let $\rho_{i,j} = \sigma_{i,j}/(\sigma_{i,i}\sigma_{j,j})^{1/2}$ denote the correlation coefficient between $z_i$ and $z_j$, $i, j = 1, \ldots, p$. Define

$$\alpha_N = \frac{p}{p + 2\log(N/r)}, \quad \mathbf{W}_N = \alpha_N \boldsymbol{\Sigma} + (1 - \alpha_N)\boldsymbol{\Sigma}\,\mathrm{diag}(\boldsymbol{\Sigma})^{-1}\boldsymbol{\Sigma}.$$

We have the following theorem.

**Theorem 4** *Suppose Assumption 1 holds and $Z \sim \mathcal{N}(\mu, \boldsymbol{\Sigma})$, $r = n/(2p)$ is an integer, there exist constants $C_1, C_2, C_3 > 0$ such that $C_1 < \lambda_p(\boldsymbol{\Sigma}) < \lambda_1(\boldsymbol{\Sigma}) < C_2$ and $\|\mu\| < C_3$, there exists a constant $0 < \rho < 1/\sqrt{2}$ such that $\max_{1 \leq i < j \leq p} |\rho_{i,j}| \leq \rho$, there exist $\epsilon_1, \epsilon_2 \in (0, 1)$ such that for sufficiently large $N$, $4r \leq N^{\epsilon_1}$, $p \leq N^{\epsilon_2}$ and*

$$(1 + 2\epsilon_2)^{1/2}|\rho| + \{(\epsilon_1 + 2\epsilon_2)(1 - \rho^2)\}^{1/2} < (1 - \epsilon_1)^{1/2}. \tag{6}$$

*Furthermore, suppose as $N \to \infty$,*

$$\frac{r}{N} \to 0 \quad \text{and} \quad \frac{p^2(\log(n))^4}{\max(n, r\log(N/r))} \to 0. \tag{7}$$

*Then as $N \to \infty$,*

$$\mathrm{E}\left\{\|\hat{\boldsymbol{\beta}}_{\mathrm{I}} - \boldsymbol{\beta}\|^2 \mid \mathbf{Z}\right\} = (1 + o_P(1))\left(\alpha_N\,\mathrm{tr}(\mathbf{W}_N^{-1}) + \alpha_N \mu^\top \mathbf{W}_N^{-1} \mu + 1\right)\frac{\sigma_\varepsilon^2}{n}.$$

**Remark 1** *If $n$ and $p$ are fixed, then the condition (6) is satisfied for sufficiently large $N$. On the other hand, if $\rho_{i,j} = 0$ for all $1 \leq i < j \leq p$, that is, the variables are independent, then the condition (6) becomes $\epsilon_1 + \epsilon_2 < 1/2$. In this case, the condition (6) holds for $n = O(N^\epsilon)$ for some $0 < \epsilon < 1/2$.*

**Remark 2** *The condition (7) is satisfied if $r/N \to 0$ and $p = O(n^{1/2-\epsilon})$ for some $\epsilon > 0$. Also, the condition (7) can be satisfied for arbitrary $n, p$ provided $N$ is sufficiently large.*

Compared with Theorem 6(i) in Wang et al. [2019], our Theorem 4 gives a more comprehensive characterization of the asymptotics of $\mathrm{Var}(\hat{\boldsymbol{\beta}}_{\mathrm{I}} \mid \mathbf{X})$. If $\mu = \mathbf{0}_p$ and $\alpha_N \to 0$, then Theorem 4 implies that

$$\mathrm{E}\left\{\|\hat{\boldsymbol{\beta}}_{\mathrm{I}} - \boldsymbol{\beta}\|^2 \mid \mathbf{Z}\right\} = (1 + o_P(1))\left(\frac{p}{2\log(N/r)}\,\mathrm{tr}(\boldsymbol{\Sigma}^{-1}\,\mathrm{diag}(\boldsymbol{\Sigma})\boldsymbol{\Sigma}^{-1}) + 1\right)\frac{\sigma_\varepsilon^2}{n}. \tag{8}$$

If we further assume that $\log(r)/\log(N) \to 0$, then the expressions (5) and (8) are equivalent. However, Theorem 4 implies that the expression (8) is not valid if $\alpha_N$ does not converge to 0.

Now we consider the special case that $Z \sim \mathcal{N}(\mathbf{0}_p, \mathbf{I}_p)$. In this case, Theorem 4 implies that if $r/N \to 0$, $n = O(N^\epsilon)$ for some $0 < \epsilon < 1/2$ and $p = O(n^{1/2-\epsilon^*})$ for some $0 < \epsilon^* < 1/2$, then $\mathrm{E}\{\|\hat{\boldsymbol{\beta}}_{\mathrm{I}} - \boldsymbol{\beta}\|^2 \mid \mathbf{Z}\}$ has convergence rate $(\alpha_N p + 1)/n$. We have

$$\alpha_N = \frac{p}{p + 2\log(2p) + 2\log(N/n)} = O\left(\frac{p}{p + \log(N/n)}\right).$$

Hence if $\log(N/n) = O(p^2)$, then $\mathrm{E}\left\{\|\hat{\boldsymbol{\beta}}_{\mathrm{I}} - \boldsymbol{\beta}\|^2 \mid \mathbf{Z}\right\} = O_P(p^2/(n(p + \log(N/n))))$ which matches (4). In this case, $\hat{\boldsymbol{\beta}}_{\mathrm{I}}$ achieves the optimal rate of sampling methods given by Theorem 2, and hence the optimal rate given by Theorem 2 is tight.

# 4 Simulation results

In this section, we conduct simulations to examine the performance of the proposed estimator $\hat{\boldsymbol{\beta}}_{\mathrm{A}}$. For comparison, the simulations also include the vanilla data averaging algorithm (abbreviated as VDA) where the full data is uniformly divided into $k := 2p$ groups in random and the observations are averaged within groups, the least square estimator based on the uniform sampling method (abbreviated as UNI), the leverage score sampling estimator (abbreviated as LEV) as described in Ma et al. [2015], the sketched least square estimator based on the subsampled randomized Hadamard transform (abbreviated as SRHT), the estimator $\hat{\boldsymbol{\beta}}_{\mathrm{I}}$ produced by the IBOSS algorithm, and the least square estimator based on the full data (abbreviated as FULL). The methods VDA, UNI, LEV, SRHT and IBOSS are instances of the sketching framework (2). For these three methods, we take $n = N/p$. For SRHT, if $N$ is a power of 2, then the sketching matrix $\mathbf{O} = (\mathbf{PHD})^{\top}$, where $\mathbf{P}$ is an $n \times N$ matrix whose rows are uniformly sampled from the standard bases of $\mathbb{R}^N$, $\mathbf{H}$ is an $N \times N$ Walsh-Hadamard matrix (see, e.g., Dobriban and Liu [2019]) and $\mathbf{D}$ is an $N \times N$ diagonal matrix whose diagonal elements are i.i.d. Rademacher random variables; and if $N$ is not a power of 2, we pad zeros to the original data to make $N$ reach a power of 2. The computation of the proposed method and the IBOSS estimator rely on certain order statistics. For these two methods, the algorithm SELECT of Floyd and Rivest [1975] is used to select the order statistics. The algorithms are implemented by C++. To be fair, for all algorithms, the estimators of $\boldsymbol{\beta}$ are solved by Gaussian elimination. The simulations are performed on a CPU with 3.30 GHz.

The statistical performance of an estimator $\hat{\boldsymbol{\beta}}$ of $\boldsymbol{\beta}$ is evaluated by the empirical mean squared error based on 100 independent replications. Specifically, the empirical mean squared error is defined as $100^{-1} \sum_{i=1}^{100} \|\hat{\boldsymbol{\beta}}^{(i)} - \boldsymbol{\beta}\|^2$, where $\hat{\boldsymbol{\beta}}^{(i)}$ is the estimator in the $i$th replication. In all simulations, the ground truth of $\boldsymbol{\beta}$ is a vector with all elements equal to 1. We consider two distributions of $\varepsilon_1$: the normal distribution $\varepsilon_1 \sim \mathcal{N}(0,1)$ and the normalized chi-squared distribution $\varepsilon_1 \sim (\chi^2(1)-1)/\sqrt{2}$. We consider the following distributions of $Z$.

- Case 1: $\{z_j\}_{j=1}^p$ are i.i.d. with uniform distribution Uniform$(0,1)$.

- Case 2: $\{z_j\}_{j=1}^p$ are i.i.d. with normal distribution $\mathcal{N}(0,1)$.

- Case 3: $\{z_j\}_{j=1}^p$ are i.i.d. with lognormal distribution, that is, $\log(z_i) \sim \mathcal{N}(0,1)$.

- Case 4: $\{z_j\}_{j=1}^p$ are i.i.d. with student $t$ distribution with 3 degrees of freedom $t_3$.

- Case 5: $Z \sim \mathcal{N}(\mathbf{0}_p, \boldsymbol{\Sigma})$, where the diagonal elements of $\boldsymbol{\Sigma}$ all equals 1 and the off diagonal elements all equals $0.5$.

- Case 6: $Z$ is distributed as a mixture of $\mathcal{N}(\mu, \boldsymbol{\Sigma})$ and $\mathcal{N}(-\mu, \boldsymbol{\Sigma})$ where $\mu$ has all 1 entries, $\boldsymbol{\Sigma}$ is defined as in Case 5, and the mixing proportions of the two component distributions are both $0.5$.

Table 2 and Tables A.1-A.3 in Supplementary Material list the empirical mean squared errors of various estimators, where the proposed method is referred to as NEW. Among the implemented methods, VDA, UNI and SRHT are data-oblivious sketching methods while NEW, LEV and IBOSS are data-aware sketching methods. It can be seen that VDA has the worst performance. This implies that the selection procedure is necessary for the proposed method. The simulation results show that UNI, SRHT, LEV have similar statistical performance. It can be seen that the proposed estimator can achieve substantial improvement over the competing sketching methods. Especially, the proposed method shows superiority when $p$ is large.

We also evaluate the computing time for various algorithms. Table 3 lists the computing time for Case 1 with $\epsilon_1 \sim \mathcal{N}(0,1)$. Results for other settings are similar. It can be seen that the proposed method is slower than VDA and UNI. Compared with VDA and UNI, however, the proposed method has significantly better statistical performance and can achieve better data reduction. Compared with IBOSS, the proposed method has a comparable computing time but a much better statistical performance. In summary, the new estimator has good performance in both speed and statistical performance.

Table 2: Empirical mean squared errors (multiplied by $10^3$) of various algorithms with $N = 8 \times 10^4$ and $\varepsilon_1 \sim \mathcal{N}(0, 1)$.

| | $p$ | NEW | VDA | UNI | SRHT | LEV | IBOSS | FULL |
|---|---|---|---|---|---|---|---|---|
| Case 1 | 50 | 182.438 | 16134 | 492.376 | 497.665 | 488.99 | 507.642 | 9.38737 |
| | 100 | 651.596 | 14751.4 | 2149.19 | 2163 | 2114.55 | 2186.94 | 19.2125 |
| | 200 | 2469.56 | 15014 | 15398.1 | 15788.1 | 14584.2 | 15159.2 | 38.0159 |
| Case 2 | 50 | 6.95906 | 1039.08 | 33.6247 | 33.5017 | 32.9702 | 27.03 | 0.624944 |
| | 100 | 21.9735 | 1039.08 | 144.342 | 144.579 | 141.629 | 125.218 | 1.28266 |
| | 200 | 69.4913 | 1034.62 | 1033.37 | 1021.28 | 1035.9 | 926.762 | 2.54829 |
| Case 3 | 50 | 5.99607 | 1008.38 | 28.2232 | 32.817 | 34.4454 | 13.635 | 0.465752 |
| | 100 | 11.6466 | 998.573 | 105.39 | 126.361 | 168.775 | 40.3553 | 1.00904 |
| | 200 | 23.0075 | 866.214 | 920.989 | 1003.37 | 982.843 | 235.444 | 1.95179 |
| Case 4 | 50 | 2.00812 | 345.559 | 12.0477 | 11.2138 | 11.6838 | 2.36205 | 0.213812 |
| | 100 | 4.66736 | 346.259 | 53.6057 | 49.8285 | 50.3076 | 7.96878 | 0.439798 |
| | 200 | 10.658 | 345.624 | 372.099 | 346.143 | 344.283 | 37.7542 | 0.856995 |
| Case 5 | 50 | 19.8421 | 2127.66 | 65.1897 | 63.4604 | 61.4904 | 59.3276 | 1.21801 |
| | 100 | 62.2782 | 1922.4 | 284.473 | 281.714 | 275.586 | 264.896 | 2.4893 |
| | 200 | 192.085 | 2015.87 | 2022.8 | 2031.46 | 1979.68 | 1973.69 | 5.00354 |
| Case 6 | 50 | 19.5601 | 1987.22 | 62.558 | 65.5263 | 62.6909 | 55.7036 | 1.22561 |
| | 100 | 60.9593 | 2022.88 | 282.951 | 286.522 | 280.046 | 263.008 | 2.47379 |
| | 200 | 194.566 | 2007.68 | 1988.91 | 2022.94 | 2000.11 | 1938.25 | 5.09764 |

Table 3: Computing time (in seconds) of various algorithms.

| $N$ | $p$ | NEW | VDA | UNI | SRHT | LEV | IBOSS | FULL |
|---|---|---|---|---|---|---|---|---|
| $8 \times 10^4$ | 50 | 0.05127 | 0.00302 | 0.00204 | 0.06567 | 0.12722 | 0.03279 | 0.05261 |
| $8 \times 10^4$ | 100 | 0.10580 | 0.00635 | 0.00332 | 0.15607 | 0.52488 | 0.07127 | 0.22732 |
| $8 \times 10^4$ | 200 | 0.21261 | 0.01618 | 0.00653 | 0.32740 | 2.14122 | 0.16388 | 0.90082 |
| $6.4 \times 10^5$ | 100 | 1.32952 | 0.05715 | 0.03292 | 1.75414 | 4.51072 | 1.16810 | 1.95327 |
| $6.4 \times 10^5$ | 200 | 2.47827 | 0.10503 | 0.05132 | 3.31281 | 17.8339 | 2.31180 | 7.56397 |
| $6.4 \times 10^5$ | 400 | 4.88818 | 0.25092 | 0.12785 | 6.86445 | 85.0793 | 4.71242 | 36.3755 |

# 5 Conclusion

In this paper, we presented a new sketching method which is based on data averaging. The computation of the proposed method can be completed within $O(Np + p^3)$ time. We proved that the proposed method can achieve a faster convergence rate than sampling methods.

In the proposed algorithm, we need to select certain order statistics of variables. This selection procedure is adapted from the IBOSS algorithm and thus allows us to compare the performance of these two methods in a fair manner. In theory, the selection of order statistics can be completed within $O(Np)$ time. However, this procedure may cost a lot of time in practice. It is interesting to investigate other selection procedures for data averaging. Also, this work focuses on the data averaging method for the linear model. It is interesting to apply the data averaging method to regularized linear models and generalized linear models. We leave these topics for possible future research.

# 6 Supplementary Material

The Supplementary Material includes additional numerical results, all proofs and codes.

## Acknowledgments and Disclosure of Funding

Wangli Xu is the corresponding author of this paper. The majority of this work was done when the first author was a postdoc in Renmin University of China. At the time the paper was accepted, the first author was with Inspur (Beijing) Electronic Information Industry Co., Ltd. This work was supported by Beijing Natural Science Foundation (No Z200001), National Natural Science Foundation of China (No 11971478), and by Public Health & Disease Control and Prevention, Major Innovation & Planning Interdisciplinary Platform for the "Double-First Class" Initiative, Renmin University of China (No. 2023PDPC), and the MOE Project of Key Research Institute of Humanities and Social Sciences(No. 22JJD910001).

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
