# OpenReview forum: "A Fast and Accurate Estimator for Large Scale Linear Model via Data Averaging"
_NeurIPS.cc/2023/Conference — NeurIPS 2023 poster_

### Official Review · Reviewer_A4mU · 2023-06-30

**Soundness:** 3 good
**Presentation:** 3 good
**Contribution:** 3 good
**Rating:** 6
**Confidence:** 1

**Summary:**

This paper studies the linear regression problem and proposes a new sketching method based on data averaging.

**Strengths:**

Please see the "questions" section.

**Weaknesses:**

Please see the "questions" section.

**Questions:**

This topic falls outside my current expertise. My review below is pretty limited.

- Comparison against some standard sketching methods such as Gaussian sketch could be valuable. Gaussian sketch is computationally expensive to apply but easier to analyze. I'm curious how the convergence rates would compare.

- I wonder if some plots would help improve the presentation. Perhaps something similar to Figure 7 of Pilanci and Wainwright [2016]?

Minor:
- There is a latex-related issue with the reference numbering in the appendix.
- typo in line 166: "no mater"

**Limitations:**

Please see the "questions" section.

---

> ### Author Rebuttal · Authors · 2023-08-09
>
>
> **Comment:**  We sincerely appreciate your valuable comments and suggestions, particularly the positive feedback regarding the proposed method.
>
> Below, we will provide a response that centers around  **questions** related to our paper. These relevant sections have been indicated with italicized text in the following.  We believe and hope that our revisions will ultimately meet your satisfaction, and that the paper will have the opportunity to be published in NeurIPS.
>
>
> **Questions:**
>
>
>
>   * *Comparison against some standard sketching methods such as Gaussian sketch could be valuable. Gaussian sketch is computationally expensive to apply but easier to analyze. I'm curious how the convergence rates would compare.*
>
>     **Reply:** Thanks a lot for your suggestion.
>     Specifically, sketching methods solve the sketched least square problem
>     \begin{eqnarray}
>     \min\_{\boldsymbol{\beta} \in \mathbb{R}^{p+1}}||\mathbf{O}^{\top} \mathbf{y}-\mathbf{O}^{\top} \mathbf{X} \boldsymbol{\beta}||^2
>     \end{eqnarray}
>     where $\mathbf{O} \in \mathbb{R}^{N \times n}$ is a sketching matrix with i.i.d.\ standard Gaussian entries.
>     Suppose $p \ll n \ll N$.
>     From Theorem 1 of Ahfock et. al. (2021),
>     under the conditions of our Theorem 5,
>     the conditional mean squared error for Gaussian sketching least square estimator has convergence rate $\sigma_{\varepsilon}^2 p / n$, which is the same as the uniform sampling method.
>     To make the computation of the least square estimator based on sketched data computed within $O(n p + p^3)$ time, we let $n \asymp N / p$.
>     In this case, the convergence rate of Gaussian sketching is $O(\sigma_{\varepsilon}^2 p^2 / N)$.
>     In this case,  the convergence rate of the proposed method, which is
>     $(1 + o_P (1))
>     \frac{p^2\sigma\_{\varepsilon}^2}{
>     2 log(2p)N
>     }
>     $,
>     is faster than Gaussian sketchign by a factor of order $\log(p)$.
>
>   * *I wonder if some plots would help improve the presentation. Perhaps something similar to Figure 7 of Pilanci and Wainwright [2016]?*
>
>     **Reply:** Thank you for your suggestion.
>     We will consider improving the presentation of our paper.
>
>
>   * *There is a latex-related issue with the reference numbering in the appendix.
>   Typo in line 166: "no mater"*
>
>     **Reply**:  Thank you for pointing out the typos.
>     We will correct typos we found,
>     and fix the reference numbering problem.
>
>
>
> **Reference**
>
> Ahfock D , Astle W J , Richardson S .Statistical properties of sketching algorithms[J].  2021.

---

> > ### Comment · Reviewer_A4mU · 2023-08-15
> >
> > Thanks for the responses. I don't have further questions.

---

> > > ### Author Response · Authors · 2023-08-17
> > >
> > > Dear Review A4mU :
> > >
> > > Thank you for your response. We appreciate your feedback and suggestion.

---

### Official Review · Reviewer_sefb · 2023-07-05

**Soundness:** 3 good
**Presentation:** 2 fair
**Contribution:** 2 fair
**Rating:** 4
**Confidence:** 3

**Summary:**

This paper considers a new estimation method for a large scale linear regression model. Specifically, the regression coefficients are estimated by least squares estimation of averaged observations for which data are partitioned via a method similar to the information-based optimal subdata selection (IBOSS) algorithm proposed in Wang et al. [2019]. The paper develops both lower and upper bounds on the mean squared error of the estimator and compares it with the existing methods theoretically as well as numerically.

**Strengths:**

- The paper proposes a new estimation method for linear regression, which is arguably one of the most important estimation problems in statistics and machine learning, although the new method shares lots of similarities with the information-based optimal subdata selection (IBOSS) algorithm proposed in Wang et al. [2019].
- The paper contains a number of interesting theoretical results.

**Weaknesses:**

- The main text of the paper does not contain any experimental results, although numerical results in the supplement are quite impressive. It would benefit the readers if the paper contains a concise summary of key numerical results in the main text.
- It is very important to compare the proposed method with the information-based optimal subdata selection (IBOSS) algorithm proposed in Wang et al. [2019] because data selection is very similar between the two methods. The paper includes lots of remarks on the IBOSS algorithm; nonetheless, it is still unclear exactly what sense they are different. It seems to me that the averaging method in the paper induces different type of weights across included observations relative to the IBOSS algorithm, but the exact link between the two methods is illusive.
- On page 25 in the supplement, a different algorithm is introduced. There are some possible typos: (i) a variant of Algorithm ??, which I presume refers to the algorithm in the main text; (ii) $r \leftarrow\left\lfloor\frac{n}{2 r}\right\rfloor$: perhaps $2p$ in the denominator. Please check them. This variant of Algorithm ?? looks much simpler but it is only introduced in the supplement. It might be useful to include this in the main text to improve the understanding of the proposed method.


**Questions:**

- The expression of "fine-grained lower bounds" in the abstract is a bit misleading because regressors are assumed to be jointly normal with mean zero vector and an identity covariance matrix. It might be better to say this limitation more explicitly in the abstract.
- Line 115: $r > 0$ is introduced but it seems that it is not defined before.
- The paper focuses on the normal regressor case. Would it be possible to extend to sub-Gaussian or other more general cases?
- In view of the remarks after Theorem 5, the proposed method has the convergence rate $p^2/(\log(2p)N)$ for the mean squared error; whereas, the convergence rate for the sampling methods is $p^2/N$. Thus, the difference is only by $1/\log(2p)$, which may not matter much when $p$ is relatively small. The abstract states that "our theoretical results show that the proposed method can achieve a faster convergence rate than the optimal convergence rate for sampling methods." However, the difference is only at the log rate and so it might be better to explicitly state that the faster rate is only up to the log factor.
- On the other hand, if we look at the numerical results in the supplement, there are huge differences between the proposed method and existing methods in terms of mean squared errors. It would be good to know where this large difference comes from. For example, is this because the subsample size $n = N/p$ in the existing methods is chosen to be too small? Some clarification would be helpful.


**Limitations:**

- One limitation is that the paper does not provide any guidance regarding how to conduct statistical inference, e.g., construction of confidence intervals. It would be helpful to provide a method for inference if possible and difficulties if not readily available.
- The supplementary material does not include replication files. It would be desirable to provide them if the paper is accepted.

---

> ### Author Rebuttal · Authors · 2023-08-09
>
> We sincerely appreciate your valuable comments and suggestions, particularly the positive feedback regarding the proposed method.
>
> Below, we will provide a response that centers around three main aspects:  **weaknesses, questions, and limitations** related to our paper. We have taken note that your feedback primarily emphasizes the presentation and explanation of the paper. We believe and hope that our revisions will ultimately meet your satisfaction, and that the paper will have the opportunity to be published in NeurIPS.
>
> **Weaknesses:**
>
>   * *The main text...*
>
>     **Reply:**  Thank you for your suggestions. Due to space constraints, all experimental results were deferred to the appendix in the original paper. In response to your feedback, we will address this by moving some content from the main paper to the appendix. If space permits, we will include Table 1 and Table 5 in the main text and include a concise summary of key experimental results.
>
>   * *It is...*
>
>     **Reply:**  Thank you for your suggestion. As you noted, the proposed method has a similar selection method as IBOSS. However, the two algorithms has essential differences which results different convergence rate. Below we elaborate the key differences. The discussion in our paper will be improved in camera-ready version.
>
>      1. The selected observations of IBOSS is only a subset of the full data. This is an inherent limitation method of sampling methods: sampling methods only select a (typically small) subsample of the full data, other samples are discarded. In comparison, the proposed averaging method selects all samples: it clusters the full data into $2p$ clusters (via the IBOSS way), and averaging within each group. Hence the averaged samples are computed from the full data, no observation is discarded.
>
>     2. The proposed method and IBOSS has different convergence rate. Theorem 6 implies that under certain technical conditions, the convergence rate of IBOSS can reach the lower bound in Theorem 2. On the other hand, Theorem 5 implies that under certain technical conditions, the convergence rate of the proposed method can break the ice and be lower than the lower bound in Theorem 2.
>
>     3. The proposed method and IBOSS have different degree of data reduction.
>     IBOSS reduces $N$ samples to $n$ samples where $n\asymp N / p$ to achieve $O(Np + p^3)$ computing time.
>     In comparison, the proposed method reduces $N$ samples to merely $2p$ observations.
>
>   * *On page 25...*
>
>     **Reply:** We deeply apologize for these typos, and will improve the presentation of our paper. The ''Algorithm ??" should be  IBOSS algorithms. The  denominator of $\frac{n}{2r}$: is  $2p$ instead of $2r$. This algorithm is coupled with IBOSS algorithm to prove Theorem 6. Since it is not related to the proposed algorithm, we decide to not put it in the main text.
>
> **Questions:**
>
>   * *The expression...*
>
>     **Reply:** Thank you for your reminder. We have  addressed this limitation in the revised paper by clarifying it in the abstract
>
>   * *Line 115...*
>
>     **Reply:** Sorry, $r = \frac{N}{2p}$ is defined latter. We will fix this mistake.
>
>   * *The paper...*
>
>     **Reply:** Theorem 1 may be generalized to non-Gaussian case.
>     We do not think Theorem 2 can be easily generalized to non-Gaussian distributions since in that case, one may need to deal with the concentration inequality of more general order statistics. Theorems 3 and 4 do not assumes $\textbf{Z}$ Gaussian. They imposes some conditions on the tail probability of $\textbf{Z}$. But the imposed conditions may not be easily verified for sub-Gaussian data. Theorems 5 and 6 rely on Gaussian assumptions. There are some $\log$ in the statement of Theorem 2 and 3. These $\log$ comes from Gaussian data. For sub-Gaussian data, the $\log$ term may not be correct.
>
>   * *In view...*
>
>     **Reply:** Following your advice, we will make clear that our theoretical results demonstrate that the proposed method can achieve a faster convergence rate, up to a $\log(p)$ factor, compared with the optimal convergence rate of sampling methods.
>
>     We would like argue a bit that the $\log(p)$ term may not be as weak as it appear.
>     In fact, from our theoretical results, the $\log(p)$ improvement can already break the lower bound of sampling methods stated in Theorem 2.
>
>   * *On...*
>
>     **Reply:**  In fact, for large $p$, the term $2\log(2p)$ in Theorem 5 may not be a small number. For example, if $p = 200$, then $2\log(2p)$ is approximately 12, which may look "huge". In this view, the good performance of the proposed method is reasonable.
>
>     Our experimental results show that compared with UNI, the proposed method has particularly good performance in Case 3 and Case 4.
>     Note that in Case 3 and Case 4, data distribution has heavy tail. Theorem 5 says nothing about the case of heavy tail.
>     Heuristically, since the proposed method relies on order statistics, it may be expected that it has excellent performance when data distribution has heavy tail. In this view, the results in Case 3 and 4 are reasonable.
>
> **Limitations:**
>
>   * *One limitation...*
>
>     **Reply:** It is convenient to use the reduced $2p$ observations to conduct statistical inference since these observations also satisfy the linear model; see the formula between line 179 and 180 of the main text. From taht formula, it can be seen that the reduced observations satisfy a regression whose error term $\bar {\varepsilon}\_j$ is a mean of independent stochastic errors. Hence from central limit theorem, it can be expected that $\bar {\varepsilon}\_j$ behaves just like a normal random error. In this view, classical statistical method for Gaussian data can be used which may produce asymptotically correct inference result.
>
>   * *The supplementary...*
>
>     **Reply:** Thank you for your suggestion. We promise that if the paper is accepted, we will open-source our code.

---

> > ### Comment · Reviewer_sefb · 2023-08-18
> > **Thank you for the rebuttal**
> >
> > I very appreciate the rebuttal by the authors. I agree that the authors' plan of restructuring the paper sounds good. Still I would like to mention that Gaussianity is a very strong assumption, as pointed by other reviewers. Also, I am not fully convinced that the $2\log(2p)$ factor is important at least in terms of theory, especially given that $p$ is relatively smaller than $n$. I am open to changing my rating but will keep it as it is now because of these concerns.

---

> > > ### Author Response · Authors · 2023-08-18
> > >
> > > Thank you for your comment and feedback.
> > >
> > > 1. Regarding non-Gaussian distributions.
> > > In experiments, it is shown that for some non-Gaussain distributions, the proposed method can have very good perform.
> > > In theory, we have pushed ourselves to prove some theoretical results regarding non-Gaussian distributions (Theorems 3 and 4).
> > > But these results may not be reader-friendly so we would like to move them to appendix.
> > > Perhaps, the general behavior of the proposed method may be too complicated to have a simple characterization.
> > > A key dificulty toward a unified theory for non-Gaussian distributions is that the proposed method involves order statistics.
> > > And the concentration behavior of order statistics may not be unified in a simple way.
> > >
> > > 2. Regarding $\log(p)$ improvement.
> > > As mentioned in your review, it can be observed that the proposed method achieves "huge" improvements over competing methods in experiments.
> > > As replied in our feedback, a log term can indeed look "huge" when the dimension is moderately large, e.g., $p=200$.
> > > So there should be no disagreement that the log term can have observable impact in practice.
> > > Now we would like to argue that *in terms of theory*, the log term may not be as weak as its first look too.
> > > In fact, based on our theoretical results, with the same order of computation time, the $\log(p)$ improvement can break the lower bound of sampling methods stated in Theorem 2.
> > > So the log improvement is not relative to a specific method, but rather to a class of methods.
> > > This class of methods include some recent method, such as IBOSS.
> > > If the $\log(p)$ term is not important, it at least proves that breaking the lower bound of Theorem 2 *is possible*.
> > > Previously, it may be unclear if there is a need to consider methods beyond sampling methods.
> > > Our theoretical work implies that considering methods beyond sampling methods can indeed have benefits.
> > > We think this information may be important in theory.
> > >
> > > We appreciate your openness to changing your rating based on these concerns. If you have any further questions or need additional clarification, please let us know.

---

> > > > ### Comment · Reviewer_sefb · 2023-08-20
> > > > **Thanks**
> > > >
> > > > Many thanks to the authors for replying to my further comments. I will try to give more thoughts on these issues during the AC-reviewer discussion period; however, at the moment, I would like to keep the initial rating.

---

> > > > > ### Author Response · Authors · 2023-08-20
> > > > >
> > > > > Dear Reviewer sefb:
> > > > >
> > > > > Thank you for your response. I truly appreciate your invaluable feedback and the thorough evaluation you have provided. Your insightful comments have greatly contributed to the reviewing process, and I sincerely thank you for investing your time and effort. I acknowledge your intention to further reflect on the discussed issues during the AC-reviewer discussion period. If you have any additional questions, please do not hesitate to reach out to us.

---

### Official Review · Reviewer_DaqC · 2023-07-07

**Soundness:** 3 good
**Presentation:** 2 fair
**Contribution:** 3 good
**Rating:** 6
**Confidence:** 2

**Summary:**

This paper gives lower bounds of the conditional mean squared error for sketching methods. They focus on least square estimator, and show that when the problem dimension is sufficiently large, the optimal error rate among all sampling reductions is achieved by uniform sampling. They also propose a sketching method based on data averaging that achieves better performance under some scenarios. Fast implementation of their proposal is also given in the paper.

**Strengths:**

This paper gives a careful analysis on the lower bound of estimation error caused by sampling methods, which is tight in many cases. They also proposes a simple yet useful sketching algorithm that has better statistical guarantees comparing to sampling based algorithms in many interesting regimes. They make the comment that the associated optimization problem could be hard to solve, and they come up with a reasonable alternative approach. This work gives nice theoretical insights on data averaging algorithms, which could help with the design of data reduction methods in practice.

**Weaknesses:**

This paper considers solely least square estimators after sketching, which could restrict its applicability in comparison to results that involve more general estimators. In the paper, they proposed a sketching method based on data averaging that is supposed to be more efficient and achieves better accuracy. However, they didn't compare their proposal with other sketching algorithms other than sampling based methods.

**Questions:**

1. Line 80 "ruduce -> reduce"
2. In the equation between line 123 and 124, X stands for the training data or the testing data?
3. Why focus on mean squared error instead of prediction error? The latter objective seems more interesting to me. If the authors have specific reason for that, it would be helpful to state it in the paper.
4. The Gaussian assumption is a bit restrictive. Does the theory work for other light-tail distributions, for example sub-Gaussian?
5. I might have missed this, but where is $\hat\beta_I$ in Theorem 6 defined?


**Limitations:**

Yes. Societal impact not applicable.

---

> ### Author Rebuttal · Authors · 2023-08-09
>
>
> **Comment:**  We sincerely appreciate your valuable comments and suggestions, particularly the positive feedback regarding the proposed method.
>
> Below, we will provide a response that centers around two main aspects: **weaknesses, questions** related to our paper. These relevant sections have been indicated with italicized text in the following.  We believe and hope that our revisions will ultimately meet your satisfaction, and that the paper will have the opportunity to be published in NeurIPS.
>
>
> **Weaknesses:**
>
> * *This paper considers solely least square estimators after sketching, which could restrict its applicability in comparison to results that involve more general estimators. In the paper, they proposed a sketching method based on data averaging that is supposed to be more efficient and achieves better accuracy. However, they didn't compare their proposal with other sketching algorithms other than sampling based methods.*
>
>     **Reply:** As you pointed out, in this work, we confine ourselves to the least square estimators after sketching, and do not consider some more general estimators such as iteration methods.
>     There are two benefits to consider such a class of estimators.
>
>     First, the considered class, although restricted, is important.
>     Indeed, within the considered class of estimators, there are not only some recent sampling and sketching methods, but also some new methods such as the proposed one.
>     Even for iteration methods, the initial point is often chosen as a sketched least square estimator which is in the considered class.
>
>     Second, the considered class allows for theoretical analysis.
>     The considered class of estimators is not too large, so it is possible prove some theorems to understand certain properties of the class, just like we did in Section 2.
>     Such results may be very difficult to obtain if one considers a larger class including more general estimators such as iterations methods.
>     Our theoretical analysis does provide insights: it shows that sampling methods may be sub-optimal.
>
>     Indeed, in the supplementary material, we provided comparisons with the sketched least square estimator based on subsampled randomized Hadamard transform (SRHT).
>     For sketched least square estimators with other sketchings,
>     the convergence property may have a similar behavior in view of the lower bound of Pilanci and Wainwright (2016).
>
>
> **Questions:**
>
>
>
>
>   1. *Line 80 ''ruduce $\rightarrow$ reduce"*
>
>       **Reply:** Thank you for pointing out the typo, and we  correct it in the updated version.
>
>   2. *In the equation between line 123 and 124, $X$ stands for the training data or the testing data?*
>
>       **Reply:** Thank you for raising this question. In lines 123-124, the symbol $X$ refers to the training data set.
>
>
>   3. *Why focus on mean squared error instead of prediction error? The latter objective seems more interesting to me. If the authors have specific reason for that, it would be helpful to state it in the paper.*
>
>       **Reply:** Mean squared error (MSE) and prediction error (within training data) can both measure the performance of an estimator.
>       MSE may be more convenient in our setting.
>       In fact, prediction error involves the data matrix $\textbf{X}$ which is random.
>       There is also randomness in $\hat \beta\_{A}$.
>       These two randomness are not independent.
>       The two dependent sources of randomness make the analysis of prediction error complicated.
>       For some other methods, either $\textbf{X}$ is assumed to be constant or $\textbf{X} (\hat \beta\_A - \beta)$ can be simplified by explicit formula of $\hat \beta\_A$.
>       For out method,  $\textbf{X}$ is random, and $\hat \beta\_A$ does not have a simple formula. So we choose MSE for convenience.
>
>
>
>   4. *The Gaussian assumption is a bit restrictive. Does the theory work for other light-tail distributions, for example sub-Gaussian?*
>
>       **Reply:**   Theorem 1 may be generalized to non-Gaussian case.
>       We do not think Theorem 2 can be easily generalized to non-Gaussian distributions since in that case, one may need to deal with the concentration inequality of more general order statistics. Theorems 3 and 4 do not assumes $\textbf{Z}$ Gaussian. They imposes some conditions on the tail probability of $\textbf{Z}$.
>       But the imposed conditions may not be easily verified for sub-Gaussian data.
>       Theorems 5 and 6 rely on Gaussian assumptions. There are some $\log$ in the statement of Theorem 2 and 3. These $\log$ comes from Gaussian data.
>       For sub-Gaussian data, the $\log$ term may not be correct.
>
>   5. *I might have missed this, but where is $\hat{\beta}\_I$ in Theorem 6 defined?*
>
>       **Reply:** $\hat{\beta}_I$ denotes the estimator of IBOSS.
>       And we add a diagram which may demonstrate IBOSS more clearly.
>
> **Reference**
>
> Mert Pilanci and Martin J. Wainwright. Iterative Hessian sketch: fast and accurate solution approximation for constrained least-squares. Journal of Machine Learning Research, 17: 1–38, 2016.

---

> > ### Comment · Reviewer_DaqC · 2023-08-17
> >
> > Thank you so much for the response! I do not have any further question.

---

> > > ### Author Response · Authors · 2023-08-17
> > >
> > > Dear Reviewer DaqC :
> > >
> > > Thanks for your response. We appreciate your comments and suggestions.

---

### Official Review · Reviewer_EUfX · 2023-07-17

**Soundness:** 4 excellent
**Presentation:** 2 fair
**Contribution:** 3 good
**Rating:** 7
**Confidence:** 2

**Summary:**

This submission studies the asymptotic estimation risk of linear regression
with various data sketching strategies.
The authors start by refining an existing lower bound to show that sketching
by uniform sub-sampling is only minimax optimal when the feature dimension
is very large. otherwise, improvement is possible and the authors prove the
previously proposed IBOSS algorithm is minimax optimal.
In order to circumvent their own lower-bound, the authors then exit the
random sketching framework to define an new method based on data averaging.
This method is shown to outperform uniform random sketching and IBOSS
both theoretically and empirically.


**Strengths:**

This submission has two main strengths:

- It completes the landscape of minimax lower bounds for random sketching
    applied to linear regression, establishing optimality of IBOSS
    in a new regime where the number of features is not too large.

- It improves on the performance of sketching methods by developing a new framework
    based on data averaging. This framework is novel, achieves faster
    (theoretical) convergence of the estimation risk, and outperforms sketching
    methods in practice. It is also comparable to IBOSS in computation time.

I cannot comment on the correctness of the theoretical developments or the
novelty of the proofs since this paper is out of my research area. Related
work seems appropriately referenced.


**Weaknesses:**

The main weakness of this paper is the presentation, which confuses the novelty
of the results and makes much of the theoretical development hard to follow.
In particular,

- The discussion is dense, mathematical, and does not give any intuition into the
    bounds proved or and the assumptions required. This makes it difficult to
    evaluate the novelty and utility of some results, in particular Theorems 3 and 4.
    No effort is made to compare the risk bounds given throughout the paper,
    although it would be very useful for Theorems 5 and 6.

- The experimental results, which make a strong argument for the utility of the
    proposed data averaging method, are all deferred to the appendix.

- The description of the data averaging method in Section 3.1a is difficult
    follow so that Algorithm 1 is more useful than the text itself.

**Questions:**

- Assumption 1: I don't see where $r > 0$ is used in the assumption.

- Theorem 1/2: How restrictive is the requirement that $Z$ be normally distributed?
    Does the result of Pilanci and Wainright (2016) require similar
    distributional assumptions on $Z$ in order to derive minimax lower bounds?

- Theorem 2: Why does $\\mathbf{E}[O (O^\\top O)^{-1} O^\\top | Z]$ being diagonal
    allow for a tighter lower bound to be derived?
    In particular, is there a chance that sketching matrices not satisfying
    this condition can out-perform the lower bound in Theorem 2?

- Displays after lines 187 and 208: Where do these expressions come from?
    The lower bound on the trace after line 208 is used to guide the development
    of Algorithm 1. Since it plays a critical role, some intuition into where
    this expression comes from is is important for the reader to follow
    the argument.

- Theorem 3: It is somewhat strange that $r = N / 2p$ is said to be an integer when
    the conditions on $p$ and $N$ require that $r \\rightarrow \\infty$.
    I suppose what you really mean is that the divergent sequences $N_i$, $p_i$
    are chosen so that $r_i = N_i / 2p_i$ is always integer?
    Is this condition simply for convenience when working with Algorithm, i.e.
    to avoid working with floor operators?

Remark 1: I think some additional comments on the tail bounds in (5)-(7)
    must be provided. Firstly, what is $\\mathcal{A}$? I only see the definition
    of $\\mathcal{A}_j$ in lines 236/237. Is this any measurable set?
    Secondly, what to do the tail bounds imply about the original dataset $Z$?
    It seems that they say something like "tail examples concentrate
    faster as dimension increases," but it's hard to tell since
    the $\\mathcal{A}$ sets also depend on $p$.
    Do these conditions make sense for general data? Can you give an example
    of a data-generating distribution for which these conditions hold, e.g.
    sub-Gaussian distributions?

Theorem 4: So, essentially $z$ follows a symmetric, isotropic distribution
    with strict conditions on the tails?
    Equation 9 implies that the tails need to be sufficiently large, but
    not too large (Equation 10); how do these conditions compare to (5)-(7)?
    Again, how they are satisfied by real data?
    I cannot understand the impact of Theorems 3/4 without understanding
    how and when these tail conditions are satisfied.

Theorem 5: Logarithmic improvement in the convergence when the number of
    features tends to infinity seems quite weak. Asymptotic regimes where
    $N \\rightarrow \\infty$ are somewhat justified since more data can be
    collected, but rarely can more features be collected.
    Without that said, I do like that the $\\log(N / n) \\ll p$ condition
    is not required, since this implies even more unrealistic asymptotics
    for $p$.
    Additionally, why not just assume $Z$ is standard normal from the start,
    rather than developing Theorems 3, 4 under awkward tail assumptions?
    Theorem 5 is much cleaner and Theorems 1, 2 already require the
    standard normal assumption.

Theorem 6: It would be nice to remind the reader the difference between
    $\\beta_I$ and $\\beta_A$ here. I think $\\beta_I$ is the IBOSS algorithm,
    but this should probably be stated in the Theorem.

Tables 1/2: It seems like every method but the one proposed experiences catastrophic failure
    when $p = 200$, performing about as well as the naive VDA method.
    IBOSS performs slightly better than competitors in Cases 3,4, but otherwise
    it also fails.
    I have two questions here: (i) Why do all other methods approach the same
    MSE for $p = 200$?
    Is this the error of an zeroth-order estimator, e.g. the mean?
    (ii) The baseline methods are all sketching-based, meaning Theorem 2 would
    lower-bound their risk, correct?
    If so, does their poor performance reflect this lower
    bound and the good performance of the proposed method relies on circumventing
    it by exiting the sketching framework?

Line 83 (Appendices): Using commas to separate methods and commas to denote
    the thousands place makes this list difficult to read. I suggest putting the
    results in a standard table with one row for improved clarity.


**Limitations:**

As mentioned in the "Weaknesses" section, the major limitation of this
paper is the presentation and quality of the writing. Specifically,

- The assumptions in Theorems 3 and 4 need to be clarified before their
    utility is clear to me. Since Theorem 5 makes a standard normal assumption
    just as Theorem 2 does, it's not clear to me why it is interesting to present
    Theorems 3/4 in the main paper when they require specific and unjustified tail bounds
    on the data distribution.

- The explanation of Algorithm should be improved. I suggest including a figure
    illustrating Algorithm 1, if possible.

- The rates in Theorems 5 and 6 should be compared and some commentary given.
    In general, more explanation of the theorems ought to be given.

- I strongly suggest reducing the technicality of the main paper and introducing
    some of the empirical results from the appendix to clarify the utility
    of the proposed averaging method. The method works well in practice,
    which should be highlighted, rather than hidden.

---

> ### Author Rebuttal · Authors · 2023-08-09
>
> We sincerely appreciate your valuable comments and suggestions. Due to space constraints, we display the discussion about non-normal cases, presentation and tables in the global rebuttal. Below, we will provide a response for main **questions and limitations**.
>
> **Questions:**
> * *Assumption 1...*
> **Reply:** In the latter part of the text, we define $r = \frac N{2p}$, and the condition $r>0$ can mathematically avoid the case $N<2p$. We would like to delete this condition and make clear where necessary that the case $N<2p$ is not considered.
>
> * *Theorem 1/2...*
> **Reply:**  The response to the first question can be seen in the global rebuttal.
> As for the second question, the minimax result of Pilanci and Wainright (2016) does not assume $Z$ is random, and hence is more general indeed. However, as we pointed out in the main text, their restriction on $\mathbf{E}_o = \mathbf{E}\left[O(O^{\top} O)^{-1}O^T|Z\right]$ is restricted, which has a different focus. As a result, the corresponding conditions for establishing the lower bound differ.
>
> * *Theorem 2...*
> **Reply:**  The diagonal condition of $\mathbf{E}_o$ restricts the class of legal sketching methods. And it is possible to obtain fine-grained result for a restricted class of methods. Technically, in the proof of Theorem 2, the diagonal condition is used to obtain
> that $\text{tr}[\textbf{X}^T\mathbf{E}_o\textbf{X}]=\sum\_{i=1}^Nd_i||X\_i||^2$.
> This equality does not generally hold if $\mathbf{E}_o$ is not diagonal. So there is a chance that sketching matrices not satisfying this condition can out-perform the lower bound in Theorem 2. Indeed, the proposed data averaging method is such a method, where corresponding $\mathbf{E}_o$ is not diagonal in general. According to Theorem 5, the proposed method can outperform the lower bound established in Theorem 2. So this is the merit of the proposed method, its existence proves it is possible to go beyond the lower bound of Theorem 2.
>
> * *Displays after lines 187 and 208...*
> **Reply:** The details for formula derivation can be seen in the PDF file attached to the global rebuttal.
>
> * *Theorem 3...*
> **Reply:**  Yes, what we really mean is that the divergent sequences $N_i,p_i$ are chosen so that $r_i=N_i/2p_i$ is always integer for convenience. We will clarify this in the main text.
>
> * *Remark 1...*
> **Reply:** Yes, $\mathcal{A}$ is a any measurable set in $p$-dimensional Euclidean space, and $\{\mathcal{A}: \Pr(Z\in\mathcal A)\leq\frac 1p\}$ represents the class of measurable set $\mathcal{A}$ satisfying $\Pr(Z\in\mathcal A)\leq\frac 1p$.
> Roughly speaking, tail bounds (5)-(7) require that the dimension $p$ is not too large and/or the tail of $Z$ is sufficiently light. They are are satisfied by standard Gaussian distribution under some conditions on dimension, see lines 429-437 in Appendix. These bounds may be satisfied by more general distributions, sub-Gaussian distributions, but to establish such results may require a lot of mathematical work. We would like to follow the suggestion and confine ourselves to the Gaussian setting.
>
> * *Theorem 4...*
> **Reply:**  Equation 9 is a weak condition. Roughly speaking, Equation 9 merely say that no $z_j$ will shrink to $\mu_j$. Equation 10 is another bound for the tail. So the conditions (5)-(7) and (10) all require the tail to be not too large. However, there seems no simple condition to unify these three conditions. Under certain conditions on dimension, these conditions are satisfied by standard normal distribution. We would like to confine ourselves to the normal distribution setting.
>
>
> * *Theorem 5...*
> **Reply:**  We agree that the logarithmic improvement is quit weak. But the merit is that it proves that the improvement is possible. Now we know there is indeed some methods with good computing time, other than samplings, may break the lower bound of Theorem 2. So if the theoretical contribution is taken into account, the present work may be meaningful.
> Previously, one of our goal is to try to make the results general and go beyond normal distribution. It turns out that the cost is greatly reduced readability.
> * *Theorem 6...*
> **Reply:** Yes, $\beta_I$ is the IBOSS algorithm.
> * *Tables 1/2...*
> **Reply:** For question (1): For competing sketching methods, we take $n=N/p$ so that the typical computing time is $O(Np+p^3)$. In this case, the error rate is of order $p^2/N$. In Tables 1 and 2, $N=8\times10^4$, $p=200$ and $n=400=2p$. For VDA method, all data are reduced to $2p$ averaged data. In this view, the result in Tables 1 and 2 is reasonable.
> For question (2): The baseline methods would be either bounded by Theorem 1 or Theorem 2. Yes, the poor performance reflect the lower bounds of Theorems 1 and 2. The newly proposed method can also be treated as a sketching method, but it violates the key conditions of Theorems 1 and 2. So it is possible to break the lower bounds of Theorem 1 and 2, resulting better performance.
>
> **Limitations:**
>
> * *The rates...*
> **Reply:**   Theorem 6 implies that, under some technical conditions, IBOSS achieves the lower bound given by Theorem 2, and hence the bound of Theorem 2 is tight. One may not expect a sampling method significantly better than IBOSS. For the typical implementation of sampling methods, one needs to take $n\asymp N/p$ to make the computing time within $O(Np+p^3)$. Theorem 5 implies that, under some technical conditions, the proposed method converges faster than sampling methods with comparable computing time, including IBOSS. Roughly speaking, the proposed method breaks the lower bound in Theorem 2. Hence better is possible.

---

> > ### Comment · Reviewer_EUfX · 2023-08-14
> >
> > Thanks for responding to my review. I read the global author response and I
> > think that the proposed changes to the organization of the text will greatly
> > improve the paper. I understand and sympathize with the desire to go beyond
> > Gaussian data; Theorems 3/4 may turn out to be valuable contributions, but
> > their complexity means they are more suited to the appendix than to the main
> > paper.
> >
> > I will discuss the submission with the other reviewers before updating my
> > score.  The changes to the manuscript also address some of the concerns put
> > forward by Reviewer sefb, so I am hopeful we can reach positive a consensus on
> > this submission.

---

> > > ### Author Response · Authors · 2023-08-17
> > >
> > > Dear Reviewer EUfX:
> > >
> > > Thank you for your review and feedback. We appreciate your kind response and value your insights in improving the overall quality of our paper through better organization of the text. We acknowledge your support for exploring beyond Gaussian data and agree that Theorems 3/4, while valuable, are better suited for the appendix due to their complexity in updated version.
> > >
> > > Thank you for engaging in discussion with the other reviewers. We hope it will lead to a positive consensus. We sincerely appreciate your valuable insight and look forward to further discussions with the other reviewers. Your updated score and feedback are highly valuable to us. Thank you for your time, effort, and contribution to improving our paper.

---

### Official Review · Reviewer_neFN · 2023-07-26

**Soundness:** 3 good
**Presentation:** 2 fair
**Contribution:** 3 good
**Rating:** 6
**Confidence:** 3

**Summary:**

This paper studies linear regression where the number of samples N is much larger than the number of predictors p (N>>p), which is computationally costly due to large N. The paper investigates lower bounds for existing sampling based methods, and proposes a novel sketching method based on data averaging which reduces the original data to a few averaged observations. Theoretical results show that the method has a faster convergence rate than the optimal convergence rate for sampling methods, and experiments show that the method reduces mean squared error over previous methods.

**Strengths:**

- The proposed method is a new sketching method for large-scale linear regression with large N, and it is original.

- The method is well-motivated from a theoretical perspective, and the lower-bounds for the proposed method under certain assumptions are better than recent sampling based methods.

- The method has $\mathcal{O}(Np + p^3)$ which is favorable when $N>>p$.

**Weaknesses:**

- There are no results in the main paper, which are quite important for a sketching paper where claims are improved computing time with reduced mean squared error. The results which are in the supplemental are not extensive, and there is only one real data example, which makes it difficult to quantify the benefits of the proposed method.
- The setup of the method is constrained to the setting with no regularization, and only the setting where $N>>p$. It might be good to add a discussion on how applicable this method is to other settings for linear models with different N and p, and with regularization.
- The presentation of the paper is quite difficult to follow.

**Questions:**

Questions:

- How does the memory usage compare for competing methods? Please add the CPU memory usage either in Table 5, or in a new table.

Suggestions:

- Minor grammatical and typographical errors should be fixed with proofreading. The citation style also does not match the Neurips style which should be fixed.

- One example for the poor presentation is that the matrix $A$ is defined after it is referenced in the text as $\beta_A$. Fixing such issues would help the paper.

- The presentation of the paper can be improved by deferring some of the related work in page 2, and details of the derivations for the theorems to the supplemental material. The generated space can be used to add numerical results, which could include some more, real data examples.

**Limitations:**

There are no clear potential negative societal impacts of the work.

---

> ### Author Rebuttal · Authors · 2023-08-08
>
> We sincerely appreciate your valuable comments and suggestions, particularly the positive feedback regarding the proposed method.
>
> Below, we will provide a response that centers around three main aspects: **weaknesses, questions, and suggestions** related to our paper. These relevant sections have been indicated with italicized text in the following. We have noticed that your feedback primarily focuses on the presentation of the paper. We believe and hope that our revisions will ultimately meet your satisfaction, and that the paper will have the opportunity to be published in NeurIPS.
>
>
> **Weaknesses:**
>
>   * *There are no results in the main paper...*
>
>      **Reply:**  Thank you for your suggestions. In the original paper, experimental results were deferred to the appendix due to the space constraint. We will consider including some experimental results in the main text.
>
>      The present paper is mainly a theory/exploratory paper. The merit of the newly proposed method is that it break the ice and proves that there indeed exists a method that can break the lower bound in Theorem 2. This proves: better is possible. This fact may be much important than the practical performance of the proposed method. While the proposed method may be far from optimal, it may shed light on the future development.
>
>   * *The setup of the method is constrained to the setting with no regularization...*
>
>     **Reply:**
>     Our results carries particular significance in the case of $N>>p$, where the proposed estimation method in the paper exhibits a faster convergence rate.
>     You give two interesting direction: regression with regularization and the case beyond $N>>p$.
>     These directions may be highly nontrivial.
>     We will consider add brief discussions on these topics in the paper.
>
>     For regularization, a straightforward method is to use the proposed method to reduce the observations to $2p$, and use the $2p$ observations to perform regularized regression.
>     One may immediately obtain some theoretical results since the reduced $2p$ observations also satisfy the linear regression model.
>     But much effort may be paid to seriously explore the fine-grained behavior of such approach.
>
>     For the case beyond $N>>p$, the methodology may be entirely different since in this case, reduction of $N$ may not be a good strategy.
>     In fact, perhaps most existing work on sketching methods for least squared problem can only work well for the case $N>>p$ since the core idea is reducing $N$.
>
>     Overall, these two directions may be good topics to consider in the future.
>
>   * *The presentation of the paper is quite difficult to follow.*
>
>      **Reply:**  Thank you for your comments. We will improve the presentation of our paper based on your following suggestions. Specifically, the presentation of the paper can be improved by deferring some of the related work in page 2, some theoretical results, and some details of the derivations for the theorems to the supplemental material. The saved space can be used to show numerical results.
>
>
>
>
> **Questions:**
>
>
>   * *How does the memory usage compare for competing methods? Please add the CPU memory usage either in Table 5, or in a new table.*
>
>     **Reply:**
>     Now we report the memory usage results.
>      The setting is as follows: the CPU frequency is 3.1 GHz; the memory is 16 GiB; the operation system is Ubuntu 22.04.2; the compiler is  gcc 11.4.0.
>      The data type is double.
>      We record the maximum resident set size (maxrss) used (in kilobytes) which is obtained via the function getrusage in sys/resource.h.
>      Since the result is slightly different in each run, we report the average result of 10 runs.
>
>     Note that the data itself takes $N \times (p+1) * 8$ bytes of memory.
>     For the case of $N = 8 \times 10^4$ and $p = 50$, the data takes 31875 KiB memory.
>     For the case of $N = 6.4 \times 10^5$ and $p = 400$, the data takes 2005000 KiB memory.
>     It can be seen that except for SRHT, the memory cost is mainly for storing the data.
>     For SRHT, we need to transform the data, so additional memory is used.
>     In our experiments, the full data is in the main memory.
>     The present work is mainly a theoretical work, and is confined to this case.
>     One interesting scenario is that the full data is too large to fit in the main memory.
>     This is a topic worth exploring in the future.
>
>
>
>     $N$ | $p$ | NEW | VDA | UNI | SRHT | LEV | IBOSS | FULL |
>      -------- | -------- | -------- | -------- | -------- | -------- | -------- | -------- | -------- |
>     $8 \times 10^4$ | $50$ | 36242 |   36516 |  35989.6 | 89050.8 | 36839.2 | 36260  | 35456 |
>     $6.4 \times 10^5$ | $400$ | 2017482 | 2019960.4 |  2012389.6 |  5300062.8  | 2025016 | 2015057.6  | 2009964 |
>
>
>
>
> **Suggestions:**
>
>
>   * *Minor grammatical and typographical errors should be fixed with proofreading. The citation style also does not match the Neurpis style which should be fixed.*
>
>       **Reply:**
>       Sorry for that!
>       We will address the presentation issue carefully.
>       Thank you for pointing out the issue of citation style.
>       We will fix it in the camera-ready version.
>
>
>    * *One example for the poor presentation is that the matrix $A$ is defined after it is referenced in the text as $\beta\_A$. Fixing such issues would help the paper.*
>
>        **Reply:**
>        Thank you for pointing out the confusion about notation.
>        The subscript of $\beta\_A$ was intended to be the initial letter of "average".
>        We will consider improving the notations in the camera-ready version.
>
>   * *The presentation of the paper can be improved...*
>
>       **Reply:** Thank you for providing detailed suggestions on the paper's presentation. We will improve the presentation in the camera-ready version.

---

### Author Rebuttal · Authors · 2023-08-09

Thank all reviewers warmly for the time you took to review and understand our paper.

Most reviewers pointed out that the presentation of the paper should be improved. The discussion is dense, mathematical. Due to space constraints, the experimental results are all deferred to the appendix. Following the reviewers suggestion for the presentation of our paper, we would like move some details of the derivations for the theorems to the supplemental material and introduce some of the empirical results from the appendix to clarify the utility of the proposed averaging method.

Specifically, the primary objective of presenting Theorems 3 and 4 in the main paper is to provide a comprehensive understanding of the theoretical framework. It is important to note that Theorem 5 assumes a standard normal distribution, which can be seen as a specific case of the more general results provided by Theorems 3 and 4. In the camera-ready version, we would like to move Theorems 3 and 4 to the appendix.
Then we can include a concise summary of key experimental results. If space permits, we will move Table 1 and Table 5 to the main text.

To make the presentation of the results clear, we plan to add Table 1 in the PDF file which summarizes the theoretical performance of the proposed method and compare it with the ideal sampling method implied by Theorem 2 and the IBOSS algorithm.
In short, the merit of Theorem 6 is that it shows that IBOSS can match the lower bound provided by Theorem 2;
the merit of Theorem 5 is that it shows that there exists a method (i.e., the proposed method) which can break the lower bound of Theorem 2.

Reviewer EUfX, Reviewer DaqC and Reviewer sefb showed that the Gaussian assumption is a bit restrictive.
Firstly, we clarify the assumptions in Theorems.
Theorem 1 may be generalized to non-Gaussian case. We do not think Theorem 2 can be easily generalized to non-Gaussian distributions since in that case, one may need to deal with the concentration inequality of more general order statistics.
Theorems 3 and 4 are more general than Theorem 5, and are used for the proof of Theorem 5. These two theorems do not assume Gaussian distribution, the cost is some involved characterizations of the tail behavior. While these two theorems may shed some lights on the behavior for non-Gaussian distribution, we actually did not rigorously give results for concrete non-Gaussian distributions.
They imposes some conditions on the tail probability of $\textbf{Z}$.
But the imposed conditions may not be easily verified for sub-Gaussian data.
Theorems 5 and 6 rely on Gaussian assumptions.
There are some $\log$ in the statement of Theorem 2 and 3.
These $\log$ comes from Gaussian data.
For sub-Gaussian data, the $\log$ term may not be correct.
Previously, one of our goal is to try to make the results general and go beyond normal distribution.
It turns out that the cost is the greatly reduced readability, which may not be a good trade-off.

---

### Decision · Program_Chairs · 2023-09-21

**Decision:**

Accept (poster)

**Comment:**

Dear Authors,

Thank you for your valuable contribution to Neurips and the ML community. Your submitted paper has undergone a rigorous review process, and I have carefully read feedback provided by the reviewers and considered the author rebuttal in detail.

This paper is focused on the asymptotic estimation risk of large-scale linear regression with various data sketching methods. The authors refine an existing estimation lower-bound and develop a new method based on data averaging. The method is shown to outperform uniform subsampling and the IBOSS method in theory and in practice. The reviewers all agree that the contribution is solid and interesting.

Given this positive assessment, I am willing to recommend the acceptance of your paper for publication.

I would like to remind you to carefully review the reviewer feedback and the resulting discussion. While most reviews were positive, the reviewers have offered valuable suggestions that can further strengthen the quality of the paper. Please take another careful look a the 'weaknesses' section of each reviewer comment. Please also review the discussions with Reviewer sefb, whose review was the most critical. I encourage you to use this feedback to make any necessary improvements and refinements before submitting the final version of your paper.

Once again, thank you for submitting your work to Neurips.

Best,
Area Chair